

# A New Look into the Impacts of Dust Radiative Forcing on the Energetics of Tropical Easterly Waves

Farnaz E Hosseinpour[1,2] and Eric M Wilcox[1]

[1] Desert Research Institute

Reno, NV 89512

[2] University of Nevada, Reno

Reno, NV 89557

Corresponding author's email address: Farnaz@dri.edu

## Keywords

Saharan Air Layer, dust, aerosol radiative forcing, wave activity, eddy kinetic energy, tropical Atlantic Ocean, African easterly jet, African easterly waves, MERRA-2, MODIS

## Abstract

Saharan dust aerosols are often embedded in tropical easterly waves, also known as African easterly waves, and are transported thousands of kilometers across the tropical



Atlantic Oceans, reaching the Caribbean Sea, Amazon Basin, and the eastern U.S.
However, due to the complexity of the African and Atlantic climate dynamics, there is still
a lack of understanding of how dust particles may influence the development of African
easterly waves, which are coupled to deep convective systems over the tropical Atlantic
Ocean and in some cases may seed the growth of tropical cyclones. Here we apply 22 years
of daily satellite observations and reanalysis data to explore the relationships between dust
in the Saharan air layer and the development of African easterly waves. Our findings show
that dust aerosols are not merely transported by the African easterly jet and the African
easterly waves system across the tropical Atlantic Ocean, but also contribute to the changes
in the eddy energetics of the African easterly waves.
The radiative forcing efficiency of dust in the atmosphere is estimated to be a
warming of approximately 20 $Wm^{-2}$ over the ocean and 35 $Wm^{-2}$ over land. This diabatic
heating of dust aerosols in the Saharan Air Layer acts as an additional energy source to
increase the growth of the waves. The enhanced diabatic heating of dust leads to the
increase in meridional temperature gradients in the baroclinic zone, where eddies extract
available potential energy from the mean-flow and convert it to eddy kinetic energy. This
suggests that diabatic heating of dust aerosols can increase the eddy kinetic energy of the
African easterly waves and enhance the baroclinicity of the region. Our findings also show
that dust outbreaks over the tropical Atlantic Ocean precede the development of baroclinic
waves downstream of the African easterly jet, which suggests that the dust radiative forcing
has the capability to trigger the generation of the zonal and meridional transient eddies in
the system comprising the African Easterly Jet and African easterly waves.



## 1 Introduction

African Easterly Waves (AEWs), also known as tropical Atlantic easterly waves, are synoptic-scale atmospheric disturbances with a preferred wavelength in the 2000-4000km range that often develop into tropical Atlantic cyclones (Dunn, 1940). The basic characteristics and behavior of the AEWs have been described in previous studies (Charney and Stern, 1962; Chang, 1993; Kiladis et al., 2006; Diaz and Aiyyer, 2013). Local heating is a dominant factor in determining the growth of AEWs over West Africa (Norquist et al., 1977), such that the presence of diabatic heating near the entrance of the African Easterly Jet (AEJ) is a favorable factor in generating AEWs (Thorncroft et al., 2008; Russell et al., 2020). The localized mid- to lower-tropospheric heating generates vortices in the vicinity of the AEJ core, which is the genesis of the AEWs (Thorncroft et al., 2008; Berry and Thorncroft, 2012). AEWs can be initiated by convective triggers over the highlands of eastern Africa and forcing from the subtropical Atlantic storm track (Cornforth et al. 2009).

Several studies have shown that AEWs are intensified in the presence of convective systems where the mesoscale convection and synoptic-scale AEWs are dynamically coupled (Kiladis et al., 2006; Hsieh and Cook, 2005&2007; Berry and Thorncroft, 2012). A large portion of tropical Atlantic cyclones and hurricanes evolve from the AEWs (Avila and Clark, 1989; Avila and Pasch, 1992; Pasch and Avila, 1994) during the boreal summer seasons, which is the season when the amplitude of AEWs peaks (e.g., Roundy and Frank, 2004).

Numerous studies addressed the dynamics of the AEWs; however, the impacts of aerosols radiative forcing on the energy of the AEWs are poorly understood. The Sahara Desert in North Africa is the largest source of dust in the world, where over sixty million



tons of dust particles (e.g., Prospero and Lamb, 2003; Lau and Kim, 2007) are lifted
annually and transported within the Saharan Air Layer (SAL) across the Atlantic Ocean
(Carlson and Prospero, 1972) and reaches the Caribbean Sea, the Gulf of Mexico, Amazon
Basin and the United States (e.g., Perry et al., 1997; Liu et al., 2008, Francis et al., 2020).
Dust particles in the SAL have a robust influence on regional and global climate through
their impacts on radiation, clouds, hydrological cycle, and atmospheric circulation
(Colarco et al., 2003; Lau et al., 2009; Wilcox et al., 2010; Kim et al., 2010). In particular,
among aerosol species, dust is known for having a strong shortwave radiative effect by
both efficiently scattering, as well as absorbing, incoming radiation and leading to a heating
of the dust layer and strong cooling of the surface (Myhre et al., 2004; Mamun et al., 2021,
Francis et al., 2022). The shortwave radiative effect is slightly counteracted by the
longwave radiative effect of dust which causes warming at the surface and cooling within
the atmosphere (Meloni et al., 2018).

14        A limited number of studies have focused on the impacts of Saharan dust plumes

on the dynamics of the AEWs (Jones et al., 2003; Ma et al., 2012; Hosseinpour and Wilcox,
2014). Jones et al. (2004) suggested that dust optical and radiative properties have
significant impacts on the AEWs. They showed that the low-level temperature anomalies
associated with the AEWs are modulated by dust radiative forcing and suggested that dust
loading in the SAL precedes the maximum geopotential height at 700-hPa by about 1-2
days. Model sensitivity studies have also shown that the intensification of AEWs can be
induced by dust (Ma et al., 2012; Grogan et al., 2019; Bercos-Hickey and Patricola, 2021;
Grogan et al., 2022). The analytical and numerical study of Grogan et al. (2016) found that
the presence of dust enhances the development of AEWs by providing a buoyancy source.





They also showed that dust can affect the propagation of AEWs by changing the wind shear
and stability of the atmosphere. Using a regional climate model coupled with a dust model,
Bercos-Hickey et al. (2017) found that Saharan dust causes AEJ to shift northward,
upward, and westward, and this results in westward expansion and the northward shift of
both the northern and southern tracks of the AEWs. Satellite observations support this
notion by showing that a similarity exists between the pattern of temperature and wind
anomalies of the AEWs and those associated with the dust outbreaks (Hosseinpour and
Wilcox, 2014).

9         Saharan dust is not the only contributor to aerosol radiative forcing over Africa

and the Atlantic Ocean. Previous studies showed that smoke transport from biomass
burning can reach up to ~ 3-5 km altitude, which is above the stratocumulus clouds over
the Sahel region, and may affect the radiation through aerosol direct and indirect effects
(Redemann et al., 2021). Biomass burning in Africa is closely related to seasonal rainfall
variability and the location of the Intertropical Convergence Zone (ITCZ); thus, the
emissions from biomass burning in North Africa occur in boreal spring and winter, when
ITCZ is south of the equator (e.g., Cahoon et al., 1992; Barbosa et al., 1999; Ramo et al.,
2020). During the boreal winter, smoke aerosols are maximized over the Sahel region
(Figure 1, Haywood et al., 2008), where the northward transport of smoke merges with
dry southward and westward transport of dust aerosols. This leads to the co-existence of
dust and smoke, as smoke is dominated on the top of the dust layer (Haywood et al., 2008).
However, during the boreal summer, biomass burning mainly occurs in South Africa,
where the air circulations transport smoke plumes toward the South-East Atlantic off-
coasts of Namibia and Angola (Zuidema et al., 2016; Cochrane et al., 2022). To study the



effects of Saharan dust aerosols on AEWs with avoiding the major impact of smoke
transport from biomass burning in South Africa, we focus our study on the region above
5° N latitude in West Africa and the eastern Atlantic Ocean in boreal Summer, where the
contribution of aerosols from biomass burning is less than 15% by mass over this region
(Matsuki et al., 2010). This study focuses on boreal summer season, because during this
season, the amplitude of AEWs peaks (e.g., Roundy and Frank, 2004), and Saharan dust
storms are active with less simultaneous transport of smoke from South Africa biomass
burning.

9          While previous studies showed the impacts of dust aerosols on climate (Ming and

Ramaswamy, 2011; Hosseinpour and Wilcox, 2014; Chen et al., 2021; Liang et al., 2021;
Grogan et al., 2022), hydrological cycle (Konare et al., 2005; Kim et al., 2010; Bercos-
Hickey et al., 2020) and cloud properties (Weinzierl et al., 2017; Haarig et al., 2019),
these elements of the climate system in this region exhibit strong variability due to AEWs.
To understand the details of interactions between dust aerosols and climate over the
Atlantic Ocean, it is essential to understand how the evolution of AEWs is determined by
both diabatic heating, as well as exchanges of eddy kinetic energy (EKE) within the jet-
wave system and how dust may contribute to the energy driving AEWs. Toward this goal,
we apply eddy energetic concepts to further analyze the relationships between dust and
the AEJ-AEWs system to gain insight into the impacts of dust aerosol radiative forcing
on the development of AEWs and the distribution of kinetic energy from the source of
instability (i.e., AEJ). Section 2 summarizes the data and methodology. Section 3
discusses the summary of results: the climatology and variability of the AEJ-AEWs
system from an energy point of view (3.1), climatology and variability of Saharan dust



aerosols across West Africa and the eastern tropical Atlantic Ocean (3.2), and the impacts
of dust on the AEJ-AEWs system (3.3). Conclusions are presented in Section 4.
**2     Data and methodology**

5       This study focuses on the relationships of Saharan dust aerosols and AEWs in

boreal summer, because during this season, the amplitude of the AEW peaks (e.g., Roundy
and Frank, 2004). We have applied the greater than 20-year time series of NASA's satellite
observations and reanalysis for the boreal summer seasons from June to August (JJA)
2000-2021 to calculate the variability of energy components of the system comprising the
AEJ, the AEWs, and aerosol radiative forcing.
**2.1    MODIS and MERRA-2 data**
To study the climatology of West Africa and the eastern tropical Atlantic Ocean,
the successor to the original Modern Era Retrospective-analysis reanalysis (MERRA;
Rienecker et al., 2008; 2011), the 3-hourly MERRA-2 (Randles et al., 1980, 2017; Buchard
et al., 1980; Gelaro et al., 2017) were used to provide more reliable assessments of climatic
and meteorological variables from 1980 to the present. The MERRA-2 reanalysis has a 3-
hourly temporal resolution and a spatial resolution of 0.5° latitude by 0.625° longitude with
72 vertical levels, extending from the surface up to 0.01-hPa.
We applied the MERRA-2 atmospheric radiative forcing that is broad band
shortwave forcing across the visible spectrum to study aerosol radiative forcing as
described in Section 2.2, as well as the meteorological variables, including wind
components, temperature, pressure and humidity from the 3-hourly MERRA-2 reanalysis



for the boreal summer (JJA) from 2000 to 2021, to calculate the eddy energetic terms of
the AEW-AEJ system as described in Section 2.3.
The reason for choosing the MERRA-2 analysis for this study is as follows: An
essential aspect of MERRA-2 is the assimilation of bias-corrected Aerosol Optical Depth
(AOD) and physical properties of aerosols from the various ground- and space-based
remote sensing platforms (e.g., Randles et al., 2017). In particular, dust is simulated in
MERRA-2 with a radiatively coupled version of the Goddard Chemistry, Aerosol,
Radiation, and Transport (GOCART; Colarco et al., 2010) aerosol model. In this manner,
the MERRA-2 system provides the best estimate of the atmosphere state historically from
the present day back to 1980.
To evaluate the MERRA-2 reanalysis with satellite observations, we used the
entire record of the  daily AOD (level 3) from two independent algorithms and well-
calibrated sensors: (I) the 550-nm Moderate Resolution Imaging Spectro-radiometer dark-
target retrieval (MODIS, MOD08_D3; Remer et al., 2021 with a 1° spatial resolution on
Terra since 2000 for the dust domains over the Atlantic Ocean, and (II) the 470-nm Deep
Blue (Sayer et al., 2019; Hsu et al., 2019) retrievals of MODIS AOD available with a 1°
spatial resolutions for the dust source regions over the land in boreal summer (JJA, 2000-
2021). The summary of the information about MODIS and MERRA-2 data product name,
variables, special and temporal resolutions are provided in Table 1.
**2.2      Aerosol radiative forcing in the atmosphere**



1       We applied the components of aerosol radiative forcing at the surface and top of

the atmosphere (TOA) from the 3-hourly MERRA-2 reanalysis datasets to calculate the
radiative forcing of dust in the atmosphere (i.e., TOA minus surface) as follows:
$$F_{aerosol} = \left(SWF_{TOA_{tot}} - SWF_{TOA_{clean}}\right) - \left(SWF_{sfc_{tot}} - SWF_{sfc_{clean}}\right) \qquad \text{Eq. (1)}$$
where $SWF_{TOA_{tot}}$ refers to the net downward shortwave radiation flux at the TOA,
$SWF_{TOA_{clean}}$ is the net downward shortwave flux at TOA under clean-sky condition,
$SWF_{sfc_{tot}}$ is the net downward shortwave flux at the surface, and $SWF_{sfc_{clean}}$ is net
downward shortwave flux at the surface under clean-sky condition.

9       To show the variability of dust, the time-longitude Hovmöller diagrams of daily

anomalies of aerosol radiative forcing are provided to represent the dust transport within
SAL across the tropical Atlantic Ocean. The daily values of radiative forcing are calculated
by time averaging the 3-hourly data. The daily anomalies of radiative forcing were
calculated with respect to the seasonal time-average of radiative forcing for each year.
These anomalies were latitudinally averaged over the latitudes of dust domains, 12-22° N.

15       To investigate the relationship between dust and the AEJ-AEWs system over the

Atlantic Ocean, we focused on the dust variability over the ocean; therefore, we consider
the location of the SAL over the tropical Atlantic Ocean, the so-called OSAL domain ,
where dust is significant from -28° to -16° E Longitude and from 12° to 22° N latitude in
the climatology of boreal summer seasons.
**2.3    Energetics of the AEJ-AEWs system**



. We applied the MEERA-2 meteorological variables as described in Section 2.1,
to calculate the eddy energetic terms associated with the distribution of kinetic energy
across the AEJ-AEWs system for the boreal summer from 2000 to 2021.
From an energy point of view, the kinematics of the atmosphere is a combination
of mean kinetic energy (MKE) of the background mean flow and eddy kinetic energy
(EKE) representing transient eddies (Lorenz, 1954). The MKE associated with the AEJ is
calculated as below, where $u$ and $v$ are horizontal components of wind and bar represents
the time-averaged over the long-term daily time series of the wind components:

$$MKE = \frac{1}{2}\left(\overline{u^2} + \overline{v^2}\right) \qquad \text{Eq. (2)}$$

The following methodology was applied to detect the 2-6 day and 6-11 day
variations associated with the AEWs (e.g., Wu et al., 2013). While many studies have
focused exclusively on 2-6 day period AEWs, several studies have found evidence that
AEWs exist on two distinct time scales of 2-6 and 6-11 day periods, as the structure of the
AEWs differs substantially between these two different time windows (Mekonnen et al.
2006; Wu et al., 2013). The time-filtering method described below was applied to
decompose EKE of the AEWs at different time-scale: 2-6 day and 6-11 day filtered
variations.
We provide the daily times series of wind components by time averaging over the
3-hourly MERRA-2 datasets. We further used the Lanczos bandpass filtering techniques
described in Duchon's (1979) study to filter the 2-6 and 6-11 day disturbances from the
daily time series of the zonal and meridional components of wind $(u,v)$. The daily
anomalies $(u',v')$ of wind components $(u,v)$ were calculated for each boreal summer season
with respect to the average of that season ($u' = u - \bar{u}$ and $v' = v - \bar{v}$; primes indicate





daily anomalies, and bars show seasonal averages). Finally, EKE was calculated as the
average of the variances of $u$ and $v$ shown as follows:

$$EKE = 1/2 \left( \overline{u'^2} + \overline{v'^2} \right)$$
Eq. (3)

The bars indicate the average over the entire JJA, 2000-2021, and the primed quantities
denote the deviation of wind components from the time-mean (daily anomalies) described
above.

7        Baroclinic conversion (BCC) is one of the most important components in the eddy

energy budget to distribute transient energy from the upstream baroclinic source across the
storm tracks downstream of the jetstream (e.g., Orlanski and Katzfey, 1991; and Chang et
al., 2002). The initiation of and the growth of the waves are significantly related to BCC,
where the transient eddies extract energy from the mean-flow through BCC (e.g., Plumb,
1986). Following the approach described in Chang et al. (2002) study, we calculated the
BCC term as below:

$$BCC = -\overline{\omega' \alpha'}$$
Eq. (4)

where $\omega$ is the rate of pressure ($\omega = \frac{dp}{dt}$) and $\alpha$ is a scale to estimate the changes in the
vertical profile of the gradient of geopotential height ($\alpha = -\frac{\partial \emptyset}{\partial p}$). We investigate BCC to
identify the locations favorable for developing EKE in the AEJ-AEWs.
**2.4    Composite analysis**

20        The composite analyses for 2-6-day and 6-11-day variations of the eddy energetics

of the AEWs were conducted for the boreal summer seasons of 22 years, 2000-2021.
Composite EKE was calculated by subtracting the EKE values associated with the lower-





quartile radiative forcing of dust from those EKE values associated with upper-quartile
aerosol radiative forcing. We find the upper- and lower-quartile aerosol radiative forcing
offshore, where the dust load is significant over the OSAL domain (rectangle in Figure 2a).
To determine the upper- and lower-quartile of aerosol forcing, the aerosol forcing over the
OSAL box is averaged at each time to create a time series of OSAL aerosol forcing. The
daily time series of aerosol radiative forcing of the grid points were spatially averaged over
the OSAL domain, which provided one single value of aerosol forcing for each individual
day in the long-term time series over the dust domain. For averaging over the OSAL
domain, area-weighted average is applied since the area of grid cells are not the same.
These time series of aerosol radiative forcing were used to select the days of the upper
quartile and the lower quartile aerosol radiative forcing for the summer season of each year.
Hence, we selected 23 days of the highest aerosol concentration (upper-quartile) and 23
days of the lowest aerosol concentration (lower-quartile) over each domain during boreal
summer of each year. From a climatology point of view, we used the upper quartile and
lower quartile of dust over 22 years of data, such that there are 506 data points to represent
the days with high values of dust concentration and 506 days with low values of dust over
each domain of study.

18       Composite EKE is provided for each grid point by subtracting the EKE values

corresponding to the upper-quartile dust days from those of the lower-quartile dust days.
Using the method explained above, the composite of the variance of zonal wind ($\overline{u'^2}$), the
variance of meridional wind ($\overline{v'^2}$), and the transient momentum fluxes ($\overline{u'v'}$) were also
calculated for boreal summer seasons, JJA, 2000-2001 (Figure 1).



## 2.5 Time-lag analysis

The time-lag analyses were conducted over each domain of study based on the following processes. Using the same methodology explained above (Section 2.4.), we used the time series of aerosol radiative forcing spatially averaged over the dust domain to select the days in the upper quartile and the lower quartile aerosol radiative forcing, such that there are 506 data points to represent the days with high values of dust concentration and 506 days with low values of dust concentration over each domain. For every 506 days of high dust concentration, we studied the time series for five days before and five days after the event to investigate the evolution of each individual dust storm. For each time series, we assigned each day of 506 days as follows: $T = 0$ for the dust-peak, $T = 1$ for one day after the dust-peak, $T = -1$ for one day before the peak of dust, and continue this for five days before and after every 506 days. We used each of these time series for 22 years and average dust radiative forcing individually for $T = 0$, $T = +/- 1$, $T = +/- 2$, $T = +/- 3$, $T = +/- 4$, and $T = +/- 5$ to gain insight into the climatology of dust evolution five days before and five days after dust peaks over each domain. We repeated the steps explained above for the 506 data points of dust in the lower quartile to provide the long-term time series of low aerosol radiative forcing over the dust domain. Finally, by subtracting the time series of the lower quartile from the upper quartile radiative forcing, we provide the composite of dust over each domain to investigate the highest variability of dust (as $T = 0$, Figure 5) and its evolution five days before and after over dust domain. Using the same methodology, we analyzed the wave activity that coincides with the upper quartile (and lower quartile) aerosol radiative forcing to investigate a possible time-lag between the dust and the




development of kinetic energy over the northern and southern track of the AEWs. The
domains selected to investigate wave activity are shown in Table 2.
**3      Summary of the results**
**3.1    AEJ-AEWs system from an energy perspective**

6         Traditional studies have used the mid-tropospheric trough and ridge from unfiltered

wind fields to diagnose the AEWs. In this manner, the AEWs trough was identified where
the meridional wind at the vertical level of the AEJ is equal to zero, indicating that the wind
shifts from northerlies to southerlies (Diedhiou et al., 1999). The existence of two distinct
tracks of the AEWs: the northern and southern tracks (e.g., Diedhiou et al., 1999; Nitta and
Takayabu, 1985; Reed et al., 1988; Wu et al., 2013) have been identified by examining the
vorticity structure of the AEWs (e.g., Carlson 1969 a&b; Thorncroft and Hodges, 2001;
Hopsch et al., 2007) and applying the reversal of the meridional gradient of potential
vorticity (e.g., Norquist et al., 1977; Pytharoulis and Thorncroft, 1999; Kiladis et al., 2006).
However, these methods are limited because of the overlapping scale of AEWs with other
phenomena and the significant amount of manual intervention required to differentiate
between synoptic-scale AEW trough axes and localized circulation centers. As a solution
to this problem, here we applied the eddy energy budget to diagnose the growth and
evolution of the AEWs.

20        Hosseinpour and Wilcox (2014) showed that the axis of the AEJ core resides at

about 600-hPa during the boreal summer; thus, here we present the results for 600-hPa,
where the activity of the AEJ-AEWs system is maximized. Figure 1a shows the mid-level
AEJ in the climatology of boreal summer. The core of the jet is zonally located from 20°



E to 30° W between the Sahel and the Sahara and spans from Africa toward the Atlantic
Ocean, where the jet axis is located at ~15° N latitude. The closed contours in Figure 1b-c
represent the MKE of the AEJ. The MKE peaks at ~12-18° N, collocated with the core of
the AEJ (Figure 1a). The long-term mean of the mid-level EKE for the 2-6-day (warm
shades in Figure 1b) and 6-11-day (warm shades in Figure 1c) bandpass filtered EKE
represents the kinetic energy of two distinct categories of the AEWs: The 2-6-day bandpass
EKE peaks offshore, downstream and along the northern side of the jet core, while the 6-
11-day bandpass EKE has a weaker signal over the northern side of the jet compared to 2-
6-day EKE. The significant signal of the 2-6-day AEWs over the tropical Atlantic implies
the significant contribution of 2-6-day transient eddies in transient disturbances over the
Ocean.
In addition, both 2-6-day and 6-11-day bandpass EKE can develop at the higher
latitudes above ~32° N toward the subtropics, which can be related to the impacts of the
westerly Rossby waves of the subtropical storm track over North Africa. These are
consistent with the previous studies, showing that after leaving the West coast of Africa,
the majority of AEWs either (1) penetrate the subtropical Atlantic Ocean via an interaction
with an extratropical trough, or (2) develop further downstream and are involved in tropical
cyclogenesis (Berry et al., 2007; Chen et al., 2008).
**3.1.1  Behaviors of transient eddies of the AEWs**
In this Section, we further investigate the characteristics of the AEWs. Figures 1d
and 1e show the climatology of transient eddies. The variance of zonal wind $(\overline{u'^2})$
represents the zonal transient eddies (Figure 1d), which peak at ~6-12° N and are elongated



downstream along the southern edge of the AEJ from approximately 15° W to 45° W.
Comparing this with Figure 1b shows that the increase of 2-6-day bandpass EKE
downstream of the jet core corresponds to the 2-6-day zonal transient eddies, whereas the
core of the 2-6-day EKE over the northern track AEWs at ~18-24° N is related to the
meridional wind variance ($\overline{v'^2}$), which represents the 2-6-day meridional transient eddies
(Figure 1e). These patterns suggest that transient eddies of the 2-6-day time-scale AEWs
are elongated both zonally and meridionally.
Figure 1f gives further information about the structure and propagation of the 2-6-
day eddies. The enhanced transient momentum flux ($\overline{u'v'}$) of 2-6-day bandpass eddies over
the northern and southern tracks of the AEWs indicates the orientation and the group
velocity of the transient eddies relative to easterly mean-flow. The positive values of the
transient momentum flux are dominant over the southern sides of the jet core, suggesting
that the southern track transient eddies propagate with a NE-SW orientation, whereas the
negative values of the transient momentum flux over the northern track suggest the NW-
SE orientation of transient eddies relative to the mean-flow. The relatively tilted
orientations of the eddies over the northern and southern track, fanning out or diverging
downstream of the jet core, are signatures of the so-called downstream development, where
transient eddy activity associated with 2-6-day AEWs is enhanced. The magnitude of the
transient momentum flux shows the 2-6-day eddies over the northern and southern tracks
of the AEWs propagate faster relative to the easterly mean-flow, whereas the values of
transient momentum flux are negligible along the AEJ axis where the mean-flow is strong.
To further investigate the behavior of the 2-6-day eddies, we discuss the baroclinic and
barotropic instability of the waves in the following Section.



### 3.1.2    Baroclinic instability of the AEJ-AEWs system

Baroclinic instability is the dynamic cause for synoptic-scale storms as a result of vertical shear of the zonal wind, corresponding to meridional temperature gradients based on the thermal wind balance (e.g., Charney, 1947; Eady, 1949). Meridional temperature gradient is also proportional to the available potential energy in baroclinic instability mechanism (Hoskins et al., 1983; Grotjahn, 2003). Baroclinic zones are defined as the favored areas for strengthening and weakening of systems, where eddies extract available potential energy from the mean-flow and convert the eddy available potential energy to EKE through baroclinic conversion (BCC) of energy (Chang et al., 2002; Orlanski and Katzfey, 1991). The changes in meridional temperature gradient also contribute to the changes in EKE of the waves (e.g., Coumou et al., J., 2015; Gertler and O'Gorman, 2019).

Previous studies showed that $\overline{u'v'}$ is an indicator of baroclinic instability at the exit region of the jet (e.g., Hoskins et al., 1983). Figure 1f represents the presence of baroclinic instability ($\overline{u'v'}$) at the northern and southern tracks of the waves downstream of the jet core, showing the development of the 2-6-day transient eddy activity downstream of the AEJ corresponds to the presence of baroclinic instability in the region where eddies can extract energy from the easterly mean-flow through baroclinic conversion (as described in the following Section). These suggest that the northern and southern tracks of the AEWs are favorable areas for the potential growth of baroclinic transient eddies as the variations in baroclinic instability tend to extract energy from the jet and convert it to eddy energy downstream of the AEJ, where the jet weakens.

We further investigated the conversion of energy through BCC by studying the fraction of the total variance of BCC (Figure 1g) attributable to variations on less than 11-



day time scales, which includes both the 2-6-day AEWs and 6-11-day AEWs. Figure 1g
shows that these variations account for a significant fraction of BCC variations over land,
where the AEJ core resides (Figure 1a), and this high fraction of BCC variance extends
offshore over the northern and southern sides of the AEJ. This is consistent with the
discussion above, suggesting the eddy activity occurs at the north and south sides of the
AEJ (Figure 1f), where the transient zonal and meridional eddies (Figures 1d-e) extract
energy from the MKE (contours in Figure 1b-c) and convert it to EKE (Figure 1b-c)
through BCC.
In the next Section, we investigated the relationships between the African aerosols
and the AEWs. Studying the time series of EKE and dust anomalies shows a similarity
between the variability of dust radiative forcing and the changes of the 2-6-day EKE over
the northern and southern tracks of the AEWs (Figures S1 and S2), suggesting a possible
impact of dust diabatic heating on the enhancement of the kinetic energy of the AEWs.
Such a relationship between dust and AEWs is also seen over each individual JJA (Figure
S3). We explore Saharan dust variability (Section 3.2) and then investigate the possible
impacts of aerosol radiative forcing of dust concentration on the energy of AEWs (Section

17 3.3).

**3.2    Saharan dust plumes- climatology and variability**
The significant dust transport from the Saharan desert across the Atlantic Ocean is
seen in the long-term mean of Saharan dust optical thickness and radiative forcing
vertically integrated over the troposphere during boreal summer (Figures 2a-c). The
inherent limitation of MODIS satellite observations is the lack of AOD data over the



highly-reflective desert regions (Figure 2a) and the Deep Blue AOD over the Ocean (Figure
2b). Because of that, based on Eq. (1) we calculated aerosol shortwave radiative forcing
from the MERRA-2 reanalysis as a complementary component (Figure 2c) to the satellite
observations. This was further examined by the scatter plots of MODIS AOD over the
Ocean ( Figure 2d) and Deep Blue over the land (Figure 2e) with respect to MERRA-2
radiative forcing, where daily data points were averaged over the oceanic and land dust
domains (rectangle in Figures 2a and 2b, respectively). This shows that MERRA-2
reanalysis is highly correlated with MODIS observations with R-values of 0.83 and 0.62,
respectively, and statistically significant with P-values less than 0.05. From a climatology
point of view, the maximum value of dust heating the atmosphere is approximately 35 $Wm^{-2}$,
, localized over the western and central Saharan Desert in JJA, 2000-2021 (Figure 2c).
The maximum value is roughly similar for each individual JJA (not shown). In addition,
the radiative forcing efficiency for atmospheric heating by Saharan dust inferred from these
scatter plots (Figures 2d-e) is roughly 20 $Wm^{-2}$ per unit AOD over the ocean and 35 $Wm^{-2}$
per unit AOD over land.

16   We investigated dust variability by studying the changes in daily radiative forcing

during dust transport across the tropical Atlantic Ocean. The longitude-time Hovmöller
diagrams of daily aerosol radiative forcing anomalies are provided for each summer from
2000 to 2021 (Figure 3). The aerosol radiative forcing is meridionally averaged over the
SAL, 12-22°N, where the dust concentration is high. The positive and negative anomalies
show the increase and decrease of aerosol radiative forcing within the SAL as dust
propagates in transient dust plumes across the tropical Atlantic Ocean. Figure 3 shows that,
on average, dust transport may reach the Caribbean Sea in less than 11 days. To investigate



the climatology of this, the fraction of total variance of dust radiative forcing was calculated
for less than 11-day and more than 11-day of dust variations during boreal summer seasons,
2000-2021 (Figures 2f-g). The variations of aerosol radiative forcing for less than 11-day
timescale variations are significant over West Africa and the eastern tropical Atlantic
Ocean and account for up to 70-80% of the total variance of aerosol radiative forcing over
these regions. In contrast, the variations of dust radiative forcing longer than 11-day are a
more significant fraction of the variance upstream, mainly over the dust sources in the
Saharan Desert.

9       We conducted similar Hovmöller analyses as above, but for MODIS observations

as a check on the dust forcing variability in the MERRA-2 reanalysis and found that the
results from MERRA-2 reanalysis were consistent with the MODIS AOD (Figure S3).
Analyzing the dust storm events from 2000 to 2021 suggests a possible relationship
between the dust transport and the variations of the AEJ-AEWs system. Our hypothesis is
that the variations of dust across the ocean during Saharan dust storms contribute to the
growth of the waves over the ocean through diabatic heating from dust radiative forcing.
To investigate this, we focus on the dust over the oceanic domain (i.e., OSAL; rectangle in
Figure 2a). The steps to study this are described in the following sections.
**3.3    Impacts of dust radiative forcing on the energy of the AEWs**

19       Previous studies have discussed the dynamics of the AEWs as summarized in the

introductory Section; however, the relationships between dust radiative forcing and the
kinetic energy of the AEWs are still unexplored. In this Section, we investigate the



relationships between dust radiative forcing of the atmosphere (TOA minus surface) and
the kinetic energy of the AEWs during the boreal summer from 2000 to 2021.
**3.3.1    Composite analysis of eddy energetics with respect to dust variability**

4        The composite analyses were conducted for the boreal summer seasons of 22 years.

The composite of the 2-6-day and 6-11-day filtered EKE (Figures 4a and 4b, respectively)
are based on the EKE values for the times that correspond to the upper-quartile dust
radiative forcing in the OSAL region (rectangle in Figure 2a) minus the EKE values of the
times correspond to the lower-quartile dust radiative forcing. The steps to calculate
composite diagrams are explained in Section 2.

10        The positive anomalies in Figure 4a show the increase of the 2-6-day EKE at the

southern track (~ 6-12°N) of the AEWs and further downstream over the northern track (~
18-24°N) coincide with the enhanced radiative forcing of dust over the offshore region.
The dipole pattern of the positive and negative anomalies may also imply a possible
southward shift of the 2-6-day EKE at the southern edge of the AEJ during high dust
concentrations. A similar dipole pattern can also be seen in Figure 4c.

16        Figure 4c shows the increase of the zonally elongated 2-6-day eddies at the southern

edge of the jet, which suggests that the strengthening of the 2-6-day zonal transient eddies
may lead to the amplification of EKE (Figure 4a) over the southern track of the waves
during dust events when aerosol radiative forcing is significant offshore. Meanwhile, the
increase of the meridional elongated transient eddies (Figure 4d) coincides with the high
concentrations of dust. Comparing this with Figure 4a suggests that during high dust
concentration in OSAL, the amplification of the 2-6-day EKE further downstream in the




northern track of the AEWs corresponds to the enhanced meridional elongated transient
eddies. While the positive anomalies of 2-6-day $\overline{u'v'}$ (Figure 4e) is a weaker signal at the
northern and southern tracks of the waves, it is still statistically significant, which shows
that the enhancement of the baroclinic instability over the northern and southern tracks of
the AEWs occurs during high aerosol radiative forcing in OSAL.

6        The negative composite along the AEJ axis at about 12-18°N (Figure 4) can be

related to the fact that the 2-6-day and 6-11-day EKE are not significant along the AEJ
axis, where the MKE and the horizontal shear of mean-flow are strong (Figure 1a-b-c). As
described in Section 3.1., the growth of transient eddies is more likely over the south and
north side of the jet, where the jet weakens and thus offers a greater chance for the
development of baroclinic AEWs (Figure 1f-g). While the negative anomaly may seem
like a reduction of eddy activity along the AEJ axis simultaneously at the time of dust
enhancement, in the next Section (3.3.2), we have evidence that the amplification of 2-6-
day EKE along the AEJ axis starts on average two days after the peak of dust offshore
(Figures 5 d-e).

16        We conducted the same composite analysis using MODIS AOD, which shows that

the results are consistent whether the MERRA-2 radiative forcing metric or the MODIS
AOD data are applied (Figure S4). Overall, these composite analyses suggest a mechanistic
relationship between the kinetic energy of the AEJ-AEWs system over the ocean and
aerosol radiative forcing during dust outbreaks in summer. The enhanced dust offshore
coincides with the strengthening of the baroclinic instability and amplification of the 2-6-
day AEWs downstream, where the jet weakens and gives a chance to strengthen the
propagation of the zonally and meridionally elongated transient eddies over the southern



and northern tracks of the waves, respectively. In the following Section, we study a possible
time lag between the occurrence of dust storms and the changes in the activity of the waves
over various domains.
**3.3.2 Time-lag between dust outbreaks and development of the AEWs**
In this Section, we investigate a possible lag between the changes of the EKE with
respect to the variability of dust radiative forcing over the OSAL. We divide the northern
track waves (18° to 24° N) and southern track (6° to 12° N) of the AEWs into two separate
regions: Eastern Atlantic (-15° to -30°E) and Central Atlantic (-30° to -45°E). We also
study the possible lag between dust in OSAL and the eddy activity downstream of the jet
core (12° to 18°N) over the eastern and central Atlantic domains (Table 2). The time lag is
investigated between composite EKE over each wave domain with respect to composite
dust radiative forcing in OSAL. The methodology for calculating time lag is described in
Section 2.
The variability of dust radiative forcing (i.e., composite for daily upper quartile
aerosol radiative forcing minus daily lower quartile aerosol radiative forcing) in Figure 5a
represents the daily variations of radiative forcing five days before and after the peak of
dust in the OSAL region for the 22 years of boreal summer seasons. This shows the
variability of dust radiative forcing associated with the dust outbreaks over the OSAL
region is significant for about six days, as it starts three days before (T = -3) and ends three
days after (T = +3) the peak of dust (T = 0), which is consistent with the timescale of the
2-6-day AEWs. Similar analyses are conducted using the upper quartile radiative forcing





only to investigate such relationships for the days with high dust concentration (Figure S5).
The results are consistent with the patterns shown in Figure 5.
Figures 5b represents the time evolution and changes in 2-6-day EKE of the
northern track AEWs further downstream over the eastern Atlantic Ocean. The changes in
EKE seem negligible at T < 0 before starting the high variations in dust in OSAL; however,
the growth of EKE occurs on average at T = 0, coinciding with the peak of dust, and then
continues growing and reaches its maximum about three days (T = +3) after the peak of
dust variations. In contrast, although a slight decrease and increase of EKE are seen
respectively before and after dust peaks, the variations of the northern track EKE over the
eastern Atlantic (Figure 5c) seem weaker compared to those further downstream.
Comparing Figure 5b with the composite analysis in Figure 4a suggests that the
enhancement of the northern track 2-6-day EKE, further downstream over the central
Atlantic, coincides with the peak of dust and is even more significant on average three days
after dust peaks in OSAL.
The negative variations of the EKE in Figures 5d and 5e at T = 0 are consistent with
the negative composite of the EKE along the AEJ axis in Figure 4. This means that the
decay of EKE along the jet axis over the Central Atlantic (Figure 5d) is initiated before
dust activity; however, the rapid growth of EKE starts on average two days (T = +2) after
the peak of dust and is maximized about three to four days (T ~ +3 to +4) after the peak of
dust in OSAL. A similar, but weaker pattern, is seen across the jet axis over the eastern
Atlantic (Figure 5e).
Figures 5f and 5g show that the changes in EKE are maintained positive before and
after dust activity. Comparing Figure 5f with Figure 5a suggests that the activity of both



dust plumes in OSAL and the southern EKE anomalies over the central Atlantic is initiated
about three days (T = -3) before dust peaks, and then amplification of EKE continues and
reaches its maximum on average two days (T = +2) after dust peaks.

4         Over the eastern tropical Atlantic (Figure 5g), the EKE variations seem negligible

during dust storms. The weaker signal of the southern track EKE variations over the eastern
Atlantic can be explained by the dynamic and energy of the AEJ-AEWs system (Figure 1),
as this is the region where the southern edge of the jet is dominant, and the MKE and
conversion of energy to EKE through BCC are significant. This suggests that while the
positive anomalies of EKE over this region coincide with the enhancement of dust in
OSAL, the influence of dust radiative forcing on changes in EKE could be weak
quantitatively compared to the amount of energy exchange between the components of the
AEJ-AEWs system at the southern edge of the jet core.

13         Comparing Figures 5b, 5d, and 5f reveals evidence of the mechanistic relationship

between variability of dust radiative forcing offshore and the changes in the 2-6-day EKE
further downstream over the Central tropical Atlantic, where the easterly flow weakens at
the exit region of the jet over the central Atlantic. On average, the peak of dust load in
OSAL occurs a few days before the amplification of the EKE downstream of the AEJ; a
similar pattern is also seen with a weaker signal over the eastern tropical Atlantic. The lag
analyses, summarized in Table 3, suggest that the peak of dust aerosols loading offshore
over the OSAL region precedes the amplification of EKE further downstream of the AEJ
over the central Atlantic Ocean. This evidence is consistent with our hypothesis on the
influence of dust radiative forcing, fueling the EKE of the 2-6-day AEWs downstream of
the AEJ over the tropical Atlantic Ocean, where tropical cyclogenesis and hurricane



activity occur. We further investigated our analyses by selecting various dust domains (e.g.,
12° to 22°N and -38° to -28°E, shown in Figure S6) and showed that our findings are
consistent regardless of the location of dust domain in SAL across the tropical Atlantic
Ocean.
**4      Conclusions**
While previous studies showed the relationship between dust transport and AEJ and
AEWs across the Atlantic Ocean (Perry et al., 1997; Liu et al., 2008; Francis et al., 2020;
Francis et al., 2021) the feedback of dust to AEJ-AEW is not well understood. A few recent
studies showed that dust affects the atmospheric dynamics of the Atlantic Ocean by
enhancing AEJ and AEW strength (e.g., Bercos-Hickey et al., 2017; Grogan et al., 2016).
However, the mechanisms of such effects are still open questions.
This study shows mechanistic relationships between the radiative forcing of dust
aerosols in SAL and the kinetic energy of the AEWs across the tropical Atlantic Ocean
using 22 years of daily satellite observations, as well as reanalysis data based on satellite
assimilation. Dust plumes across the Atlantic are not merely transported by AEJ-AEWs
system but also contribute to increasing the kinetic energy of the baroclinic AEWs through
diabatic heating. The enhanced dust contributes to an increase in meridional temperature
gradients, which leads to an increase in baroclinicity and amplification of the EKE of the
AEWs.
The radiative forcing efficiency of dust in the atmosphere is a heating of roughly
20 Wm$^{-2}$ per unit AOD over the ocean and 35 Wm$^{-2}$ per unit AOD over land. This agrees
with in-situ measurements (Soupiona et al., 2020) and regional climate modeling (Saidou



Chaibou et al., 2020) of Saharan dust radiative forcing. This radiative forcing of dust
aerosols in the SAL contributes to the diabatic heating of the atmosphere in the regions
where the increase in temperature gradients leads to the growth of baroclinic waves through
the conversion of energy to EKE in the AEJ-AEWs system. Outbreaks of high dust
concentrations in the SAL coincide with the growth of the meridionally elongated 2-6-day
transient eddies over the northern track of AEWs (~18-24°N) and zonally elongated eddies
over the southern track of AEWs (~6-12°N). This leads to amplifying the EKE of the
AEWs, particularly at the exit region of the AEJ, where the MKE and the horizontal shear
of mean-flow are weakened. This offers the chance for downstream development of the
AEWs, associated with enhanced dust. The dust-induced enhancement of AEW through a
buoyancy source was shown by Grogan et al. (2016), albeit with a different methodology
(i.e., analytical and regional modeling analyses). In addition, our results agree with a case
study of the Saharan dust event by a regional climate model (Bercos-Hickey et al., 2017)
that showed that Saharan dust causes AEW to shift northward and expand westward.

15         The growth of the baroclinic transient eddies, and the corresponding EKE of the 2-

6-day AEWs, is amplified at the exit region of the AEJ, on average, two to four days after
the enhancement of dust upstream in the OSAL region. Our findings show that dust activity
precedes the amplification of EKE, suggesting that the diabatic heating from dust radiative
forcing can fuel the development of the AEWs. This mechanistic impact of dust radiative
forcing onto AEW development is consistent across the tropical Atlantic Ocean. This study
further supports a cause-and-effect hypothesis between dust radiative forcing and transient
wave dynamics that may be tested in sensitivity studies with dynamical climate models to
explore further the cause and effect of such relationships.





## Acknowledgments

This work is supported by the NASA Interdisciplinary Science Program through grants #NNX11AF21G and #NNX14AH95G. Special thanks to Drs. Hans Moosmuller and Naresh Kumar for their comments that contributed to the improvement of this manuscript. We also appreciate the anonymous reviewers for their constructive comments.

## Data availability

MERRA-2 aerosol, radiation, and meteorological datasets can be obtained from https://disc.gsfc.nasa.gov/datasets. MODIS AOD retrievals are accessible through https://modis.gsfc.nasa.gov/data/dataprod/mod04.php. Numerical codes developed to conduct data extraction, analysis, and visualization will be provided upon request.

## Author contributions

FH and EW originated this study. FH formulated, developed and implemented the codes, and analyzed the results with inputs from EW. FH drafted the paper, and EW provided edits and revisions.

## Competing interests

The authors have no competing interests.




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

**Table 1** MODIS and MERRA-2 data information applied in this study

| Dataset | Product Name | Variables | Spatial Resolution | Temporal Resolution | Data Reference |
|---|---|---|---|---|---|
| MODIS | MOD08_D3 | 550-nm AOD, Deep-blue AOD | 1°×1° | daily | Platnick (2015) |
| MERRA-2 | M2I6NPANA | U, V, T, H | 0.5°×0.625° | 3-hourly (averaged to daily) | GMAO (2015a) |
| | M2T1NXRAD | $SWF_{TOA_{tot}}$, $SWF_{TOA_{clean}}$, $SWF_{sfc_{tot}}$, $SWF_{sfc_{clean}}$ | 0.5°×0.625° | 1-hourly (averaged to daily) | GMAO (2015b) |
| | M2I3NPASM | Omega | 0.5°×0.625° | 3-hourly (averaged to daily) | GMAO (2015c) |

**Table 2** The coordinates of domains of transient changes across the tropical Atlantic
Ocean:

| AEW domains | | |
|---|---|---|
| Description | Central Atlantic | Eastern Atlantic |




| Northern track waves | 18° to 24°N -45° to -30°E | 18° to 24°N -30° to -15°E |
|---|---|---|
| Downstream of jet-axis | 12° to 18°N -45° to -30°E | 12° to 18°N -30° to -15°E |
| Southern track waves | 6° to 12°N -45° to -30°E | 6° to 12°N -30° to -15°E |

3 **Table 3** Summary of lag analyses showing AEWs evolution before and after dust peaks in

4 OSAL:

| Downstream development of eddy activity – Central Atlantic | | | |
|---|---|---|---|
| | Before Dust-peak | Simultaneously at Dust-peak | After Dust-peak |
| Northern track AEWs | T < 0 Negligible changes in EKE | T = 0 EKE starts increasing | T = +3 Max EKE |
| Along the AEJ axis | T < 0 Negligible changes in EKE | T = 0 Decrease of EKE | T = +2 EKE starts increasing<br><br>T ~ +3 to +4 Max EKE |
| Southern track AEWs | T = -3 EKE starts increasing | T = 0 Increase of EKE | T = +2 Max EKE |



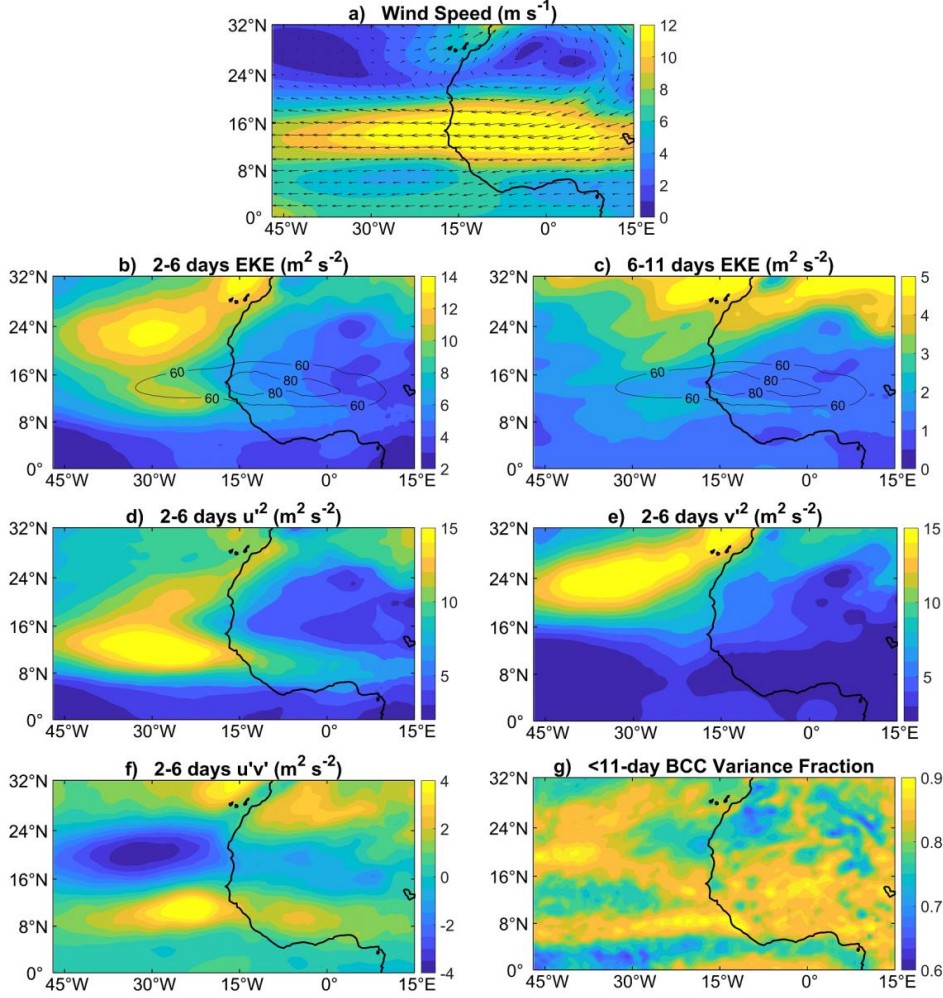

**Figure 1** (a) Long-term mean of 600-hPa wind speed (ms$^{-1}$) from MERRA-2 reanalysis over JJA, 2000-2021. (b) Same as (a) but for 2-6-day bandpass filtered EKE (m$^2$s$^{-2}$) at 600-hPa. (c) Same as (b) but for 6-11-day bandpass filtered EKE. (d) Same as (b) but shows the 2-6-day variance of zonal wind, $\overline{u'^2}$, (m$^2$s$^{-2}$). (e) Same as (b) but shows the 2-6-day variance of meridional wind, $\overline{v'^2}$, (m$^2$s$^{-2}$). (f) Same as (b) but for the 2-6-day filtered transient momentum fluxes, $\overline{u'v'}$, (m$^2$s$^{-2}$). (g) Fraction of less than the 11-day variance of 600-hPa Baroclinic Conversion (BCC) with respect to the total variance of BCC in JJA, 2000-2021.



**Figure 2** (a) Long-term mean of 550-nm aerosol optical depth (AOD) from the MODIS over JJA, 2000-
2021. (b) Same as (a) but for 470-nm MODIS deep-blue AOD. (c) Same as (a) but for aerosol shortwave
radiative forcing (Wm⁻²) in the atmosphere (TOA minus surface) from the MERRA-2 reanalysis. (d)



Relationship between MODIS AOD and MERRA-2 radiative forcing for JJA, 2000-2021. Each data point
shows daily data averaged over the OSAL region (rectangle in 2a). The results are statistically significant
with P-value < 0.05. (e) Same as (d), but for MODIS deep blue AOD over the land (rectangle in 1b). (f)
Fraction of variations of less than 11-day for the variance of aerosol radiative forcing with respect to the total
variance using the long-term mean of aerosol radiative forcing in the atmosphere (TOA minus surface) from
the MERRA-2 reanalysis over JJA, 2000-2021. (g) Same as (f) but for variations of more than 11-day.



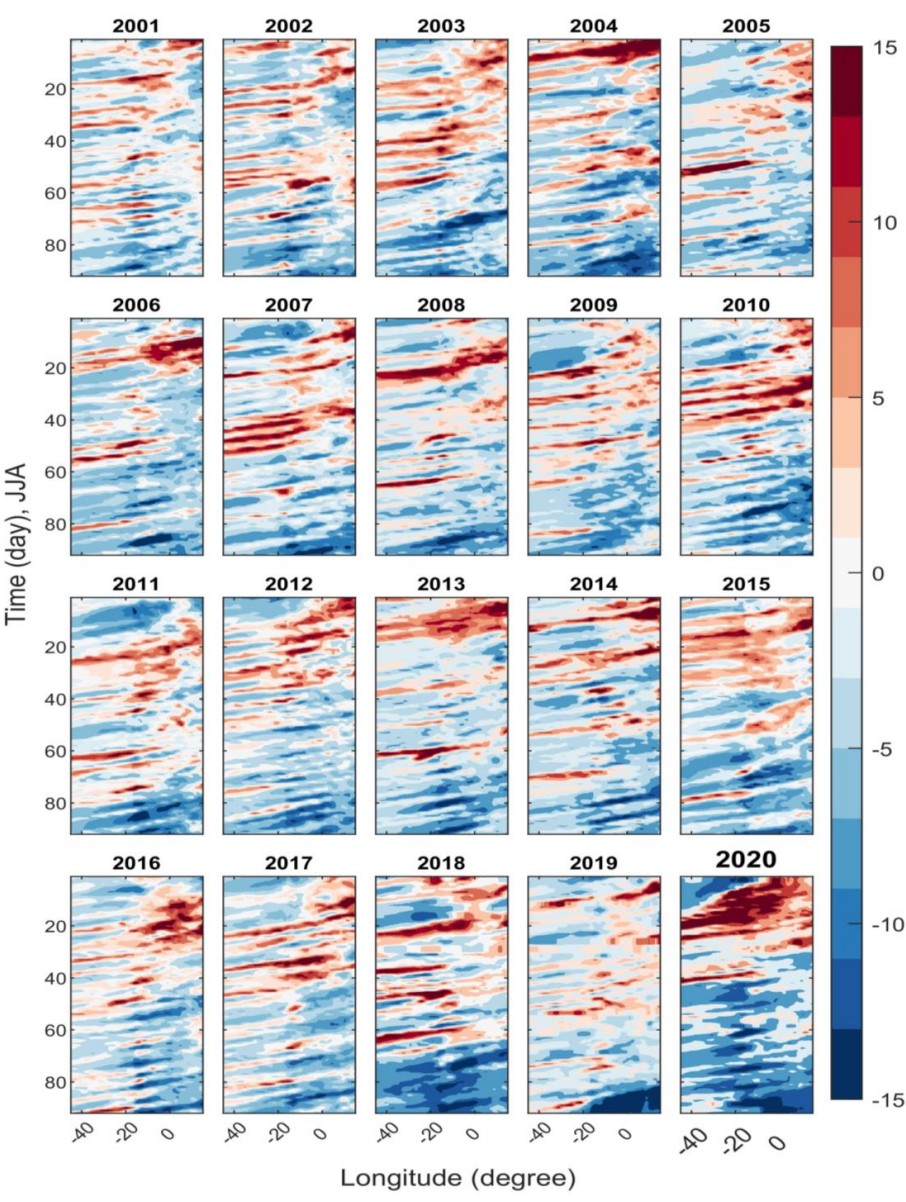

**Figure 3** Time-longitude Hovmöller diagrams of aerosol radiative forcing daily anomalies (Wm⁻²) using the
MERRA-2 reanalysis for all individual boreal summer seasons, JJA from 2000 to 2021, meridionally
averaged (12-22° N) over the OSAL domain (rectangle in Figure 2a). Daily anomalies of aerosol radiative
forcing are calculated with respect to the seasonal time average of radiative forcing for each year.

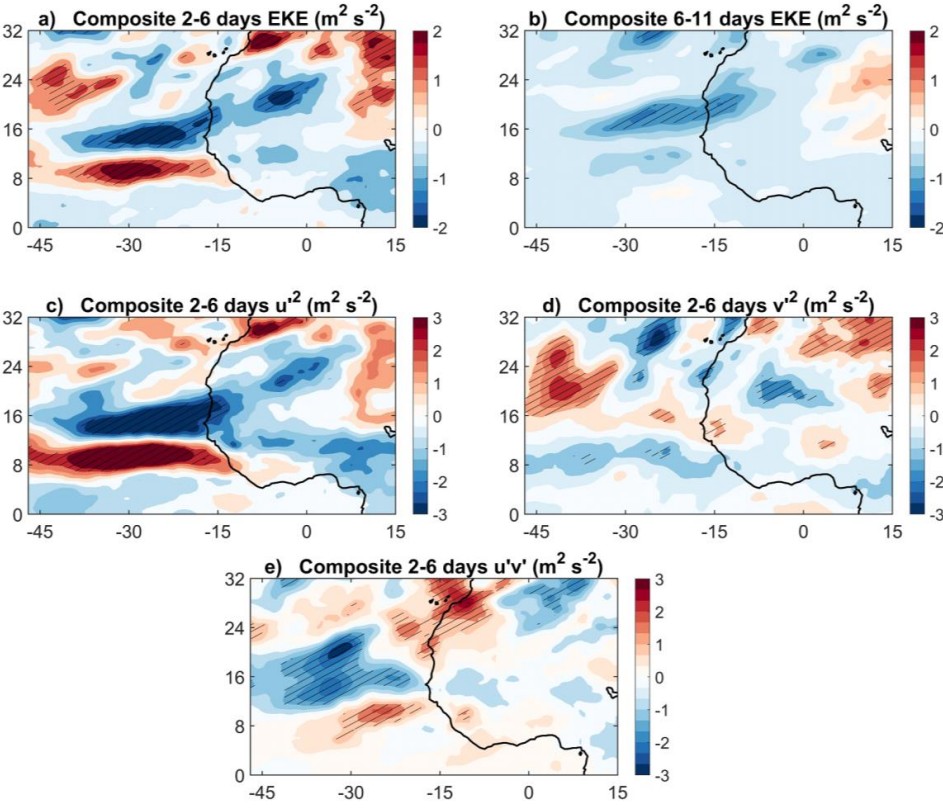

**Figure 4** (a) Composite 600-hPa 2-6-day filtered EKE (m²s⁻²) values for the times corresponding to the upper quartile aerosol radiative forcing minus the EKE values of the times corresponding to the lower quartile aerosol radiative forcing over the OSAL domain (rectangle in Figure 2a). The calculations are conducted using the MERRA-2 reanalysis for JJA, 2000-2021. (b) Same as (a) but for 6-11-day filtered EKE (m²s⁻²). (c) same as (a) but for the 2-6-day variance of zonal wind, $\overline{u'^2}$, (m²s⁻²). (d) As in (a) but for 2-6-day the variance of meridional wind, $\overline{v'^2}$, (m²s⁻²). (e) Same as (a) but for the 2-6-day filtered momentum fluxes, $\overline{u'v'}$, (m²s⁻²).



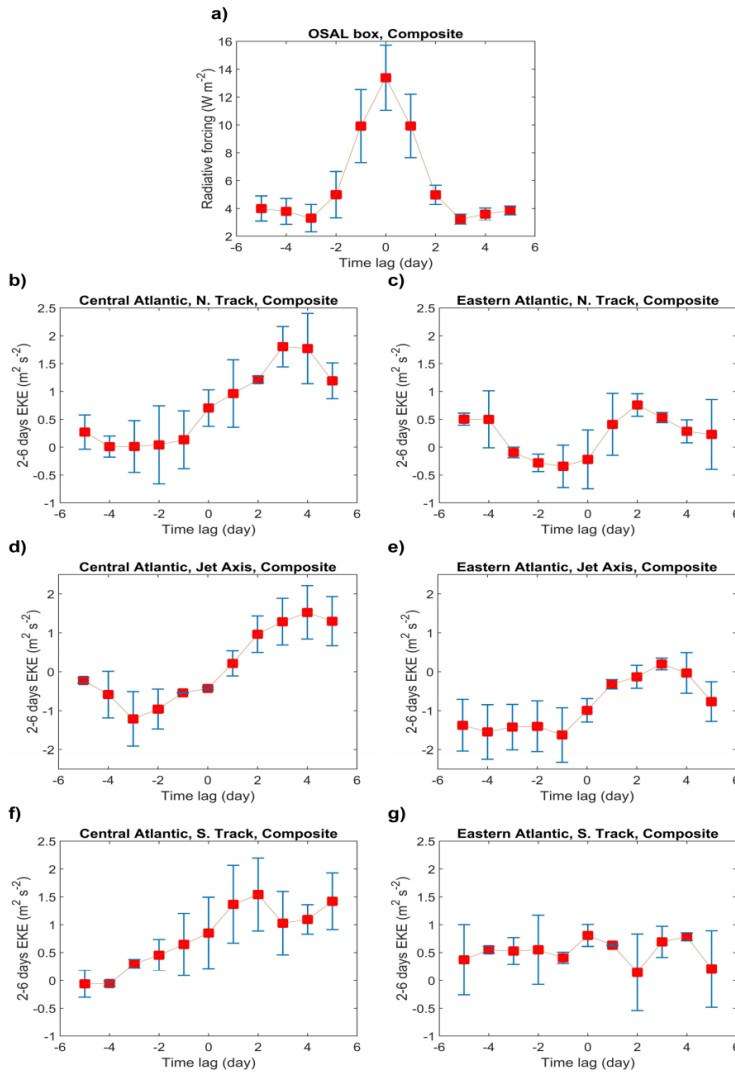

**Figure 5** (a) Daily time series of composite aerosol radiative forcing for the days in the upper quartile
minus those days in the lower quartile radiative forcing, spatially averaged over the OSAL domain
(rectangle in Figure 2a). T = 0 is assigned for the days with the highest variability of aerosol radiative
forcing in the OSAL. T = +/- 1, T = +/- 2, T = +/- 3, T = +/- 4, and T= +/- 5 are assigned for five days
before and five days after of each individual dust event, averaged over for 22 years, JJA, 2000-2021. (b)
Same as (a) but for the composite 2-6 day filtered EKE at 600-hPa, spatially averaged over the northern
track AEWs in the central Atlantic (18° to 24°N, -45° to -30°E). (c) Same as (b) but for the eastern Atlantic
(18° to 24°N, -30° to -15°E). (d) Same as (b) but spatially averaged over the domain, downstream of the
AEJ in the central Atlantic (12° to 18°N, -45° to -30°E). (e) same as (d) but for the eastern Atlantic (12° to
18°N, -30° to -15°E). (f) same as (b) but spatially averaged over the southern track of the AEWs in the
central Atlantic (6° to 12°N, -45° to -30°E). (g) Same as (f) but for the eastern Atlantic (6° to 12°N, -30° to
-15°E). The domains of the wave activity are listed in Table 3.