# Peer review of "A New Look into the Impacts of the Dust Radiative Effects on the"

_EGUsphere, 2022_

## Referee Comment (RC2)

**Manuscript Summary**

This study explores the relationships between dust in the Saharan air layer and the development of African easterly waves across the tropical Atlantic Ocean using 22 years of daily satellite AOD observations, as well as reanalysis MERRA-2 data based on satellite assimilation.

**General Comments**

**Introduction:**

Most references were dated. There is a lack of essential updated references in the manuscript. I found plenty of articles related to the topics described in this manuscript that were not reviewed. Here are a few examples:

1. Francis et al. (2021).The dust load and radiative impact associated with the June 2020 historical Saharan dust storm. https://doi.org/10.1016/j.atmosenv.2021.118808
2. Meloni et al., (2018). Determining the infrared radiative effects of Saharan dust: a radiative transfer modeling study based on vertically resolved measurements at Lampedusa. https://acp.copernicus.org/articles/18/4377/2018/
3. Bercos-Hickey et al., (2017). Saharan dust and the African easterly jet–African easterly wave system: Structure, location, and energetics. https://doi.org/10.1002/qj.3128
4. Konare et al., (2005). A regional climate modeling study of the effect of desert dust on the West African monsoon. https://doi.org/10.1029/2007JD009322.
5. Grogan et al., (2017). Effects of Saharan Dust on the Linear Dynamics of African Easterly Waves. https://doi.org/10.1175/JAS-D-15-0143.1.
6. Grogan et al., (2019). Structural Changes in the African Easterly Jet and Its Role in Mediating the Effects of Saharan Dust on the Linear Dynamics of African Easterly Waves. https://doi.org/10.1175/JAS-D-19-0104.1
7. Bercos-Hickey et al., (2019). Structural Changes in the African Easterly Jet and Its Role in Mediating the Effects of Saharan Dust on the Linear Dynamics of African Easterly Waves. https://doi.org/10.1175/JAS-D-19-0104.1
8. Francis et al. (2020). The Atmospheric Drivers of the Major Saharan Dust Storm in June 2020.  https://doi.org/10.1029/2020GL090102

**General Body:**

1. The manuscript, in its current state, has room for improvement. For example, flow,  the order of the figures' numbering, and lack of tables to help the audience to understand the results the authors intend to disseminate.
2. There is no discussion about their results (especially section 3 and conclusions) with updated research (For example, the above references). The manuscript needs to be improved and justified through discussions of novel peer-reviewed publications.
3. The intention of using satellite data is essential. But, the co-authors need to investigate each algorithm's appropriate use more. For example, I can't entirely agree with using

MERRA-2 AOD or MODIS Level 3 AOD retrievals. MERRA-2 Reanalysis and MODIS Level 3 retrievals underestimate/overestimate aerosol loading for dust and smoke. On the other hand, level 2 satellite aerosol retrievals characterize better aerosol loading. There is information in the manuscript that is not entirely correct. I encourage co-authors to investigate more about these aerosol products.

4. This manuscript needs better organization. It reads disorganized and rushes in the necessary information.

**Specific Comments**
**Introduction:**

1. There was no mention of the significant African aerosol components or the African dust and fire seasons.
2. It should be mentioned when the Biomass and Dust season overlap.

**Data and Methodology:**

1. While the temporal domain is stated, the spatial domain is hard to find. Also, I am unsure if the averaging method is per grid size or for the entire square domain. How many grids are contained in those squares? It would be helpful to state all those details to understand the methodology further.
2. It would be helpful to create a Methodology section for data manipulation of AOD. While AOD was vaguely mentioned, unfortunately, there is no explanation of how the co-authors intended to use it. In addition, there is no information about wavelength or the retrieval collection used.
3. Page 6; Lines 1-4: No relevant. Provide information just about MERRA-2.
4. Page 6; Line 21: dated reference: Please update for:
    a. Remmer et al. (2021). The Dark Target Algorithm for Observing the Global Aerosol System: Past, Present, and Future. https://doi.org/10.3390/rs12182900
5. While MERRA-2 Reanalysis is cited, it's unclear at this point what variables were used. MERRA-2 contains multiple aerosol weather variables assimilated. It would be helpful to create a table with the variables employed in the analysis.
6. Page 6; Line 22: he Deep-blue algorithm is mentioned but never cited. A few useful references:
    a. Sayer et al. (2019). Validation, Stability, and Consistency of MODIS Collection 6.1 and VIIRS Version 1 Deep Blue Aerosol Data Over Land. https://doi.org/10.1029/2018JD029598.
    b. Hsu et al. (2019). VIIRS Deep Blue Aerosol Products Over Land: Extending the EOS Long-Term Aerosol Data Records.
7. Page 7: Lines 1-4. The selection of the studied period states:"… *This study focuses on boreal summer months, JJA, from 2000 to 2021 because during this season, the amplitude of AEWs peaks (e.g., Roundy and Frank, 2004), and Saharan dust storms are active without simultaneous transport of smoke aerosols from biomass burning as*

*observed during the fire seasons…."* I suggest all co-authors study the NASA ORACLES campaign.

    a. Redemann et al., (2020). An overview of the ORACLES (ObseRvations of Aerosols above Clouds and their intEractionS) project aerosol–cloud–radiation interactions in the southeast Atlantic basin. https://acp.copernicus.org/articles/21/1507/2021/

    b. Cochrane et al., (2019). Above-cloud aerosol radiative effects based on ORACLES 2016 and ORACLES 2017 aircraft experiments. https://amt.copernicus.org/articles/12/6505/2019/

    c. Cochrane et al., (2021). Biomass burning aerosol heating rates from the ORACLES (ObseRvations of Aerosols above Clouds and their intEractionS) 2016 and 2017 experiments. https://amt.copernicus.org/articles/15/61/2022/

8. I am skeptical that in 20 years of the study, the atmosphere would not have a mix of smoke and dust on specific episodes. But unfortunately, there is no methodology from the co-authors to warranty the presence of dust.

9. Page 8: Line 6. It would be helpful to the reader to start refereeing the figures in order. For example, referencing Figure 2 confused me and made me go back several times to ensure I did not miss Figure 1.

10. Section 2.3: The weather variables used in this study were finally introduced here. However, this format is very disorganized because this section supposes to be about the calculation of MKE instead of introducing MERRA-2 variables or algorithms.

11. Page 8: Lines 9-11. The introduction of GOCART should not have been explained in this section, which is about the formulation of MKE.

    a. Randles et al. (2017). The MERRA-2 Aerosol Reanalysis, 1980 Onward. Part I: System Description and Data Assimilation Evaluation. https://journals.ametsoc.org/view/journals/clim/30/17/jcli-d-16-0609.1.xml

    b. Goddard Chemistry, Aerosol, Radiation, and Transport model (GOCART)

12. For sections 2.3, 2.4, and 2.5, are the averages per grid cell or for the entire domain?

**Summary of the results**

1. Page 17. Lines 1-9. Deep-blue has an ocean retrieval for Levels 2 and 3 data and for VIIRS. Please, edit accordingly. This would not be a challenge using level 2 data for this manuscript. Check page 127 https://atmosphere-imager.gsfc.nasa.gov/sites/default/files/ModAtmo/L3_ATBD_C6_C61_2019_02_20.pdf

2.

3. Page 17. Line 17: How did you calculate aerosol shortwave radiative forcing from the MERRA-2? It would be nice to reference Eq. 1

4. Have you compared MERRA-2 AOD with the satellite data? While correlation factors are high and p values are low, the magnitude in the color bars and gradients of the data in the maps are different. I want to remind co-authors that two datasets can have high

correlation factors with high biases. My concerns are related to the ocean gradient specifically.

5. Page 19. Lines 1-3. I encourage you to show these results in an appendix.
6. Section 3.3 states previous studies have discussed the development of the AEWs, but I did not find any reference or comparison to other studies in sections 3.3.1 and 3.3.2.

**Conclusions**

Conclusions in this manuscript are not reinforced by any peer-review publication.

**Data availability**

Page 26. Line 9: Use MODIS AOD retrievals instead of observations. This study did not use direct observations from MODIS.

**Figures**

Figure 2.a. Did you mean MODIS Dark-Target AOD (550 nm) ? Did you use 500 nm instead of 550 nm?
https://atmosphere-imager.gsfc.nasa.gov/sites/default/files/ModAtmo/L3_ATBD_C6_C61_2019_02_20.pdf
Figure 2.b. Did you mean MODIS Deep-Blue AOD (550 nm). I encourage consistency in figs 2 a and b.

---

## Author Comment (AC1)

**Response to Reviewers**

We thank the reviewers for their detailed and constructive comments on our paper. We have responded to all of the comments and changed the revised manuscript accordingly. In the following response, the original reviewer comments appear in blue, and our responses appear in black.

**Reviewer #1**

Review of "A New Look into the Impacts of Dust Radiative Forcing on the Energetics of Tropical Easterly Waves" by Hosseinpour and Wilcox.

This paper explores the relationship between dust activities and the developments of the African easterly wave (AEW) system over the Atlantic Ocean. Their hypothesis is that dust aerosols enact radiative/diabatic heating, which results in an increased meridional temperature gradient in the baroclinic zone over the Atlantic. Since this zone is where "eddies extract available potential energy from the mean-flow and convert it to eddy kinetic energy," the authors suggested that increased dust activities over the region, therefore, results in increased eddy kinetic energy (EKE) of the AEW. In addition, they showed that maximum dust radiative effect could be associated with increased EKE downstream 2-3 days later.

Because a broader understanding of the influence of African dust and the AEW system is useful to advance our understanding of the region, this study is relevant to the scientific community, and it fits into the broad scope of the journal.

While previous studies have established that aerosol radiative effect can result in changes in near-by dynamical systems, such relationships have often been shown in the absence of other confounding factors. For example, the authors cited Jones et al. 2004, who used the difference between first guess (which does not include appropriate parameterizations to account for the radiative and microphysical effects of mineral dust) and the analysis (which incorporate satellite radiance that includes dust impacts) in NCEP to estimate the impacts of dust on the AEW system. This differentiation between dust-laden and dust-free analysis is important to bring out the effects of dust on a dynamical system that is strongly coupled to its variability. Such a potential caveat to the conclusion of the analysis is not considered in this manuscript. Specifically, the author failed to also show that high values of AOD also occur when the AEW-AEJ system is strong, and therefore, the so-called "mechanistic relationship" described in the manuscript may be ascribed as a mere coincidence. Therefore, a better case of dust resulting in changes in EKE can be made if a similar signal, as described in the manuscript, results in days with high AOD but weak/low mid-tropospheric winds (e.g., AEJ). Authors should include such analysis and revise the conclusion accordingly.

We thank reviewer#1 for their constructive comments and suggestions. To answer the above comment and if we understand the reviewer's comment correctly, with MERRA-2 alone we cannot compare a dust-free circulation to a dusty circulation. This is something that can be explored with model experiments and our current work will be followed up with a modeling simulation of with and without dust. To clarify this, we added a few sentences in both the methods and conclusion Sections:

Page 8, Lines 9-19: "*It is important to note that the dust and the circulation are fully coupled in MERRA-2. Using such an empirical tool, it is not possible to directly compare a complete representation of the circulation without dust to the circulation with dust. However, the benefit of using MERRA-2 is that it offers a more realistic representation of the circulation than an unconstrained model because of the data*"

*assimilation. It is our intention with this study to evaluate the empirical relationships between the dust radiative effect and the energetics of AEWs in a reanalysis constrained by observations, which can be compared with the results of a follow-on examination of a controlled experiment in an unconstrained atmospheric general circulation model comparing simulations with dust radiative effects to simulations without dust radiative effects."*

*Page 28, Lines 12-18: "The data assimilation in MERRA-2 provides a realistic representation of circulation, while one caveat of using MERRA-2 alone is that it is not possible to compare a dust-free circulation to a dusty circulation. The empirical relationships apparent from this study will be examined in a follow-on study of atmospheric general circulation model simulations with and without the dust radiative effect to further explore the hypothesis linking dust radiative effects to AEW dynamics."*

**Minor Comments**

- 2/2 – (page/line(s)): "complexity of…..climate dynamics"?
  We modified the sentence on Page 2, lines 2-3:*"… due to the complex climate dynamics of West Africa and the eastern tropical Atlantic Ocean…"*

- 2/11: here and everywhere else after this – Is it radiative "forcing" or radiative "effect"? There is a substantial difference between the two.
  We agree that our calculation is aligned with the aerosol radiative "effect", so it has been changed throughout the paper.

- 5/19: remove "the", and say exactly the number of years you used.
  Done; Page 7, Line 7: *"…We used a 22-year time series of …"*

- 6/2: "original version". Is there a different version than what is publicly available? Please clarify.
  We had used version 2 of MERRA data, known as MERRA-2 data. Nevertheless, we removed "… original …" to prevent confusion on Page 7, Lines 14-15: *"… the Modern Era Retrospective-analysis reanalysis (MERRA; Rienecker et al., 2008; 2011), the 3-hourly MERRA-2 …"*

- 6/12: What are these "physical properties"? Other than AOD, I don't know what aerosol "physical properties" that are assimilated into MERRA-2. Please clarify what exactly is assimilated and why MERRA-2 is used here, instead of other reanalysis datasets.
  "physical properties" has been removed from Page 8, Line 5: *" … (AOD) from the various ground- and space-based remote sensing platforms …"*

- 7/3-4: "….and Saharan dust storms are active without simultaneous transport of smoke aerosols from biomass burning as observed during the fire seasons. ….". Are you suggesting that there is NO smoke emission from west/North Africa during the summer season at all? I believe this statement is incorrect. Also, what is the "fire season"?
  We adjusted the wording in many places to refer to aerosols generally, rather than dust specifically, in recognition that dust may not be the sole aerosol species in the North Atlantic Ocean region. We have added references in the introduction section of the paper to clarify the sources and transport of dust and smoke aerosol species. We have expanded the discussion in the Introduction Section in the revised manuscript with more discussion about the production of smoke aerosols, in addition to dust aerosols. This includes additional references to key literature on the production and transport of smoke aerosols.

- 7-8/section 2.2: Please see the comment above, radiative "forcing" is different from radiative "effect". I believe you are referring to the radiative effect here, and perhaps everywhere else in the manuscript, and not forcing.

We changed "radiative forcing" to "radiative effect" throughout the manuscript.

- 8/6: Figure numbering should start from 1 and order according to when they first appear in the manuscript.

Done.

- 8/10: Dust is not "assimilated" from the GOCART model. Please re-write this sentence, and please be accurate when you make sentences.

We rephrased it on *Page 8, Line 4: " … essential aspect of MERRA-2 is the assimilation of bias-corrected Aerosol Optical Depth (AOD)…"*

The assimilation of AOD is the phrase used in the MERRA-2 peer-reviewed paper: Randles et al., 2018.

- 8/11: remove "best"

Done; Page 8, Line 8: " *… the MERRA-2 system provides an estimate of the atmosphere state …"*

- 8/12: remove "the"

Done; Page 10, Line 7: *"We used MERRA-2 meteorological variables …"*

- 8/11-15: You do not "applied….to calculate…", you "used….to calculate". Please re-write this sentence.

Done; Page 9, Lines 8-10: *"We used the …. to calculate…"*

- 8/20: Did you first average your 3-hourly data before doing the daily long-term average? Please clarify how **exactly** the datasets are processed for this case and every other case in your methodology.

For clarification, we added the following sentences on Page 10, Lines 9-12: *"While the MERRA-2 data is 3 hourly, we averaged them for each day to be consistent with the daily temporal resolution of MODIS AOD data. We provided daily MERRA-2 data to apply them for the calculation of the eddy energetics terms."*

- 9/1: "following methodology " what methodology?

This has been modified on Page 10, Lines 19-20: "*To detect the 2-6 day and 6-11 day variations associated with the AEWs, we used the methodology following Wu et al. (2013)."*

- 17/6 and 19/3: "(not shown)". These figures should be shown in the supplementary.

We added Figures S1, S2 and S3 to address the reviewers' comments:
Page 18, Line 22 *"…northern and southern tracks of the AEWs (Figures S1 and S2),…"*
Page 19, Line 1-2: *"Such a relationship between dust and AEWs is also seen over each individual JJA (Figures S1, S2, and S3)."*
Page 20, Line 19-22: *"We conducted similar Hovmöller analyses as above, but for MODIS observations as a check on the dust forcing variability in the MERRA-2 reanalysis and found that the results from MERRA-2 reanalysis were consistent with the MODIS AOD (Figure S3)."*

Regarding the statement *"the maximum value is roughly similar for each individual JJA (not shown)"*: Since this is not the main point of the manuscript, we believe adding more than 20 figures may not add further information to the paper; thus, we deleted this sentence from the manuscript.

- 21/19-20: Please show these boxes in Fig. 4. It appears the boxes are significantly bigger than the defined region of EKE in Fig. 4. Can you explain what determines the selected regions?

We mentioned the exact coordinates of these boxes in Table 2. As described in Table 2, we selected three latitudinal bands: northern track waves, downstream of the jet-axis, and southern track waves, over the ocean, where the eddy energetic components and their composite with respect to dust variability are strong. The latitudes of the boxes have been selected over the areas with strong shades in Figure 4; thus, adding the boxes will make it noisy and distract the readers from its main message. We longitudinally divided the region into central and eastern Atlantic (Table 2) since the time lag analysis showed the impacts of aerosols on wave activity may vary as waves propagate westward across the tropical Atlantic; thus, we longitudinally divided the domains into two regions: central and eastern tropical Atlantic.

- 25/1: A direct relationship between dust and temperature gradient that connects to the baroclinicity is not made in this analysis. So why are the authors making this claim?

The relationship between baroclinicity and temperature gradient has been described in Section 3.1.2, Page 17, Lines 11-20. Also, Hosseinpour and Wilcox (2014) showed the relationship between dust and temperature gradients. We added the reference to the following paragraphs:
Page 27, Lines 6-8: *"The enhanced dust contributes to an increase in meridional temperature gradients (Hosseinpour and Wilcox, 2014), which leads to an increase in baroclinicity and amplification of the kinetic energy of the AEWs."*
Page 27, Lines 12-15: *"This radiative effect of dust aerosols in the SAL contributes to the diabatic heating of the atmosphere in the regions (Hosseinpour and Wilcox, 2014) where the increase in temperature gradients leads to the growth of baroclinic waves through the conversion of energy to EKE in the AEJ-AEWs system."*

- 25/20: "…cause-and-effect hypothesis…." Remove "cause-and-effect "

"Cause-and-effect" has been removed from Page 28, Line 10. We also included the following paragraph on Page 28, Lines 13-18: *"The data assimilation in MERRA-2 provides a realistic representation of circulation, while one caveat of using MERRA-2 alone is that it is not possible to compare a dust-free circulation to a dusty circulation. The empirical relationships, apparent from this study will be examined in a follow-on study of atmospheric general circulation model simulations with and without the dust radiative effect to further explore the hypothesis linking dust radiative effects to AEW dynamics."*

**Reviewer#2**

**Manuscript Summary**
This study explores the relationships between dust in the Saharan air layer and the development of African easterly waves across the tropical Atlantic Ocean using 22 years of daily satellite AOD observations, as well as reanalysis MERRA-2 data based on satellite assimilation.

**General Comments**
**Introduction:**

Most references were dated. There is a lack of essential updated references in the manuscript. I found plenty of articles related to the topics described in this manuscript that were not reviewed. Here are a few examples:

1. Francis et al. (2021).The dust load and radiative impact associated with the June 2020historical Saharan dust storm. https://doi.org/10.1016/j.atmosenv.2021.118808
2. Meloni et al., (2018). Determining the infrared radiative effects of Saharan dust: aradiative transfer modeling study based on vertically resolved measurements atLampedusa. https://acp.copernicus.org/articles/18/4377/2018/
3. Bercos-Hickey et al., (2017). Saharan dust and the African easterly jet–African easterlywave system: Structure, location, and energetics. https://doi.org/10.1002/qj.3128
4. Konare et al., (2005). A regional climate modeling study of the effect of desert dust onthe West African monsoon. https://doi.org/10.1029/2007JD009322.
5. Grogan et al., (2017). Effects of Saharan Dust on the Linear Dynamics of African EasterlyWaves. https://doi.org/10.1175/JAS-D-15-0143.1.
6. Grogan et al., (2019). Structural Changes in the African Easterly Jet and Its Role inMediating the Effects of Saharan Dust on the Linear Dynamics of African Easterly Waves.https://doi.org/10.1175/JAS-D-19-0104.1
7. Bercos-Hickey et al., (2019). Structural Changes in the African Easterly Jet and Its Role inMediating the Effects of Saharan Dust on the Linear Dynamics of African Easterly Waves.https://doi.org/10.1175/JAS-D-19-0104.1
8. Francis et al. (2020). The Atmospheric Drivers of the Major Saharan Dust Storm in June2020. https://doi.org/10.1029/2020GL090102

**General Body:**
1. The manuscript, in its current state, has room for improvement. For example, flow, the order of the figures' numbering, and lack of tables to help the audience to understand the results the authors intend to disseminate.

We thank reviewer#2 for their constructive comments and suggestions.

- Figures are discussed in Section 3 (Summary of the results), and the order of figures' numbering is as follows
    - Figure 1 in Section 3.1,
    - Figures 2 and 3 in Section 3.2;
    - Figure 4 in sub-Section 3.3.1;
    - Figure 5 in sub-Section 3.3.2.

To prevent confusion, we removed mentioning "rectangle in Figure 2a" from the last paragraph section 2.2 and only addressed the location using the lon and lat.

There are 3 Tables in the revised version of the manuscript:
- **Table 1** is added to summarize the MODIS and MERRA-2 data information in the revised version of the manuscript. This table includes product name, variables, spatial resolution, temporal resolution, and reference to data products. The references related to these datasets are included in the list of References.
- There were two Tables in the manuscript: **Tables 2 and 3**, which summarize the results.

2. There is no discussion about their results (especially section 3 and conclusions) with updated research (For example, the above references). The manuscript needs to be improved and justified through discussions of novel peer-reviewed publications.

The reviewer is correct to suggest that we should have made more effort to explain how our results relate to some of the prior literature on African Easterly Waves. Rather than do this in section 3, where we report the details of the results of our study, however, we have chosen to do this in the **Conclusion Section** of the paper, where it seemed more natural to summarize the results and reflect on where they fit within the context of the prior literature.

3.The intention of using satellite data is essential. But, the co-authors need to investigate each algorithm's appropriate use more. For example, I can't entirely agree with using MERRA-2 AOD or MODIS Level 3 AOD retrievals. MERRA-2 Reanalysis and MODIS Level 3 retrievals underestimate/overestimate aerosol loading for dust and smoke. On the other hand, level 2 satellite aerosol retrievals characterize better aerosol loading. There is information in the manuscript that is not entirely correct. I encourage co-authors to investigate more about these aerosol products.

Without a reference from the reviewer for their conclusion that level 2 MODIS data are superior to either MERRA2 or MODIS Level 3, we found it rather challenging to determine exactly how the reviewer has come to this conclusion. Nevertheless, our study primarily focuses on the variability in aerosols over the tropical Atlantic Ocean, rather than absolute values of AOD. We have chosen to primarily use the radiative forcing of aerosols in the atmosphere as our desired metric because there is a strong diabatic heating signature in the atmosphere due to aerosols that we have previously found to be associated with positive temperature perturbation of African Easterly Waves (Hosseinpour and Wilcox 2014). This is a metric that a satellite alone is not capable of quantifying, but MERRA2 does provide. In order to build confidence in the use this MERRA2 metric, our paper shows that the atmospheric radiative forcing in MERRA2 is well-correlated with the MODIS AOD, which is not totally unexpected, since MERRA2 assimilates the MODIS AOD. Nevertheless, we have gone further in the revisions provided here to show that the top-line results of our study on the relationships between African Easterly Waves and aerosols are consistent whether computed using MERRA2 atmospheric radiative forcing or MODIS AOD. So even if the reviewer is correct that MERRA2 underestimates aerosols and MODIS level 3 overestimates aerosols, our results appear to be robust to these differences, perhaps because, as noted above, we are largely investigating the variations in aerosols and their relation to wave activity.

To return to the original point that the reviewer argues; that level 2 MODIS data are more reliable than either MERRA2 or MODIS level 3 data, we posed this question to the principal investigator of the MODIS aerosol products to find out if he could point us to a reference for this argument. He could not provide any evidence for a bias in level 3 data relative to level 2 data, but also conceded that because the level 3 data are aggregated on 1 degree lat/lon grids, it is not a simple task to do a validation study of these data (R.C. Levy, NASA GSFC, personal communication January 19, 2023). There is evidently no ground truth measurement at comparable scale to compare with. Finally, we disagree with the suggestion by the reviewer that performing a multi-decadal analysis of aerosol data over an entire ocean basin with level 2 data is simple to accomplish. Indeed, NASA goes to no insignificant level of effort to produce reliable, quality assured software and processing procedures to aggregate terabytes of level 2 data into level 3 data products. Our goal is to seek insights into the influence of aerosols on weather and climate using these data products. Producing a new level 3-like average from terabytes of level 2 data is outside of the scope of this study, and as noted above, is likely unnecessary given the consistency of our results using either MERRA2 or MODIS level 3.

4. This manuscript needs better organization. It reads disorganized and rushes in the necessary information.

In the revised version of the manuscript, we added **Section 2.1** to mainly explained the MODIS and MERRA-2 information. **Table 1** is also added to summarize this information in Section 2.1.
In addition to Table 1, there are two more tables (Tables 2 and 3) to summarize the results.

**Specific Comments**

**Introduction:**
1. There was no mention of the significant African aerosol components or the African dust and fire seasons.

We have expanded the discussion in the **Introduction Section** in the revised manuscript with more discussion about the production of smoke aerosols, in addition to dust aerosols. This includes additional references to key literature on the production and transport of smoke aerosols.

2. It should be mentioned when the Biomass and Dust season overlap.

We have included new discussion in the introductory section on the transport of smoke aerosols to the North Atlantic region, including references to prior literature that have quantified the fraction smoke aerosol contribution to total aerosol amount in the North Atlantic region during the boreal summer months. Matsuki et al. (2010) estimate that smoke accounts for less than 15% of the aerosol in this region during the summer season.

**Data and Methodology:**
1. While the temporal domain is stated, the spatial domain is hard to find. Also, I am unsure if the averaging method is per grid size or for the entire square domain. How many grids are contained in those squares? It would be helpful to state all those details to understand the methodology further.

The information about averaging are provided in the Methodology Section 2:
**Section 2.4**: *"The daily time series of aerosol radiative forcing of the grid points were spatially averaged over the OSAL domain, which provided one single value of aerosol forcing for each individual day in the long-term time series over the dust domain*"
We added another sentence for further clarification *"To determine the upper- and lower-quartile of aerosol forcing, the aerosol forcing over the OSAL box is averaged at each time to create a time series of OSAL aerosol forcing."*
In addition, for averaging a variable over a specific box, area-weighted average is applied, because the area of grid cells is not the same. Area-weighted averaging is a common method of averaging variables on grids and is used extensively in climate science. This is considered by default as most papers may not mention it.
Nevertheless, we added the sentence below to clarify this in Section 2.4.
*"For averaging over the OSAL domain, area-weighted average is applied since the area of grid cells are not the same."*

Section 2.4 "Composite analysis"
2. It would be helpful to create a Methodology section for data manipulation of AOD. While AOD was vaguely mentioned, unfortunately, there is no explanation of how the co-authors intended to use it. In addition, there is no information about wavelength or the retrieval collection used.

- **Table 1** is added in the revised version of the manuscript to summarize MODIS and MERRA-2 data information, described in subsection 2.11. The information about **product name (retrieval collection), list of variables, wavelength, spatial resolution, and temporal resolution** along with references to the data are included in Table 1.

**-** In the revised version of the manuscript, **Section 2.1** is dedicated to the information of MODIS and MERRA-2 data and also summarized in Table 1 to address the reviewers' comments about Methodology Section.
-The title of the sub-section 2.1 is slightly modified in the revised version to **"MODIS and MERRA-2 data"**, showing the information about MODIS and MERRA-2 data is organized under this sub-section.

3. Page 6; Lines 1-4: No relevant. Provide information just about MERRA-2.

This has been deleted in the revised manuscript.

4. Page 6; Line 21: dated reference: Please update for: a. Remmer et al. (2021). The Dark Target Algorithm for Observing the Global Aerosol System: Past, Present, and Future. https://doi.org/10.3390/rs12182900

The data reference is added to the reference in the revised manuscript.

5. While MERRA-2 Reanalysis is cited, it's unclear at this point what variables were used. MERRA-2 contains multiple aerosol weather variables assimilated. It would be helpful to create a table with the variables employed in the analysis.

In the revised version of the manuscript, the list of MERRA-2 variables is provided **in Section 2.1** and summarized in **Table 1** as well.

6. Page 6; Line 22: he Deep-blue algorithm is mentioned but never cited. A few useful references: a. Sayer et al. (2019). Validation, Stability, and Consistency of MODIS Collection 6.1 and VIIRS Version 1 Deep Blue Aerosol Data Over Land. https://doi.org/10.1029/2018JD029598.
b. Hsu et al. (2019). VIIRS Deep Blue Aerosol Products Over Land: Extending the EOS Long-Term Aerosol Data Records.

Sayer et al. (2019) and Hsu et al. (2019) are added to reference the MODIS deep blue algorithm.

7. Page 7: Lines 1-4. The selection of the studied period states:"… *This study focuses on boreal summer months, JJA, from 2000 to 2021 because during this season, the amplitude of AEWs peaks (e.g., Roundy and Frank, 2004), and Saharan dust storms are active without simultaneous transport of smoke aerosols from biomass burning as observed during the fire seasons…."* I suggest all co-authors study the NASA ORACLES campaign. a. Redemann et al., (2020). An overview of the ORACLES (ObseRvations of Aerosols above Clouds and their intEractionS) project aerosol–cloud–radiation interactions in the southeast Atlantic basin. https://acp.copernicus.org/articles/21/1507/2021/

b. Cochrane et al., (2019). Above-cloud aerosol radiative effects based on ORACLES 2016 and ORACLES 2017 aircraft experiments. https://amt.copernicus.org/articles/12/6505/2019/
c. Cochrane et al., (2021). Biomass burning aerosol heating rates from the ORACLES (ObseRvations of Aerosols above Clouds and their intEractionS) 2016 and 2017 experiments. https://amt.copernicus.org/articles/15/61/2022/

The reviewer is correct to note that we cannot be totally certain that the aerosols causing positive aerosol radiative forcing in the atmosphere over the Tropical Atlantic Ocean are purely dust aerosols. For this reason, we have generalized the language throughout the paper to refer to aerosols and aerosol forcing generally, rather than dust and dust forcing. We have also added additional discussion about the production of smoke aerosols and their potential transport to the dust outflow region, including references to the ORACLES literature under **Introduction Section**. Since the focus of this paper is aerosol radiative forcing over the tropical Atlantic Ocean and subtropical North Atlantic and their relationship to the associated African Easterly Waves, we have chosen not to expand the bounds of the analysis to include the later months of the calendar year or the ORACLES region in the South Atlantic Ocean. While the presence of smoke aerosols in that region is important for the clouds and climate of the South Atlantic Ocean, we believe that these are separate scientific questions that are beyond the scope of this particular study.

The paragraph referred by the reviewer is removed from last part of the Methodology Section as it is described in the methodology Section. added to introduction in the revised version of the manuscript.

8. I am skeptical that in 20 years of the study, the atmosphere would not have a mix of smoke and dust on specific episodes. But unfortunately, there is no methodology from the co-authors to warranty the presence of dust.

We have added references in the introduction section of the paper to clarify the sources and transport of dust and smoke aerosol species. We have adjusted the wording in many places to refer to aerosols generally, rather than dust specifically, in recognition that dust may not be the sole aerosol species in the North Atlantic Ocean region. The goal of this paper is to explore the relationships between aerosols and African Easterly Wave dynamics in a part of the world that is known to be dominated by dust aerosols. Our discussion of the relevant literature is now clear about relative abundance of smoke aerosols in relation to dust and total aerosol amount over the North Atlantic Ocean.

9. Page 8: Line 6. It would be helpful to the reader to start refereeing the figures in order. For example, referencing Figure 2 confused me and made me go back several times to ensure I did not miss Figure 1.

- To avoid confusion, we removed referring to Figure 2 in the last paragraph Section 2.2 of the old version of the manuscript, as the domain has been addressed by lon and lat, both in the old version and the revised version of the manuscript; thus, it doesn't seem required to refer to Figure 2 in Section 2.2.
-The figures are all explained in order in the next section (Section 3). The order of figures' numbering is as follows: Figure 1 in Section 3.1, Figures 2 and 3 in Section 3.2, Figure 4 in sub-Section 3.3.1, and Figure 5 in sub-Section 3.3.2.

10. Section 2.3: The weather variables used in this study were finally introduced here. However, this format is very disorganized because this section supposes to be about the calculation of MKE instead of introducing MERRA-2 variables or algorithms.

- We moved the information about MERRA-2 weather variables to **Section 2.1 (MODIS and MERRA-2 data)** for organization in the revised version of the manuscript.

11. Page 8: Lines 9-11. The introduction of GOCART should not have been explained in this section, which is about the formulation of MKE. a. Randles et al. (2017). The MERRA-2 Aerosol Reanalysis, 1980 Onward. Part I: System Description and Data Assimilation Evaluation. https://journals.ametsoc.org/view/journals/clim/30/17/jcli-d-16-0609.1.xml

- To avoid confusion, we removed the lines referred to GOCART as the information about MERRA-2 and GOCART has been provided under Section 2.1.
- Randles et al. (2017) has been included in Section 2.1 in the revised manuscript.

b. Goddard Chemistry, Aerosol, Radiation, and Transport model (GOCART)

The long name of GOCART has been mentioned in Section 2.1

12. For sections 2.3, 2.4, and 2.5, are the averages per grid cell or for the entire domain?

The daily time series of aerosol radiative forcing of the grid points were spatially averaged over the OSAL domain, which provided one single value of aerosol forcing for each individual day in the long-term time series over the dust domain. For averaging over the OSAL domain, area-weighted average is applied since the area of grid cells are not the same. This information is available in the manuscript under **Section 2.4** (composite analysis). For **Sections 2.3 and 2.5** these are time-averaging to provide the maps as explained in these Sections in the manuscript.

**Summary of the results**
1. Page 17. Lines 1-9. Deep-blue has an ocean retrieval for Levels 2 and 3 data and for VIIRS. Please, edit accordingly. This would not be a challenge using level 2 data for this manuscript. Check page 127 https://atmosphere-imager.gsfc.nasa.gov/sites/default/files/ModAtmo/L3_ATBD_C6_C61_2019_02_20.pdf

The addition of VIIRS data to MODIS would indeed add additional aerosol information. We have chosen, as noted above, to use the MERRA2 atmospheric aerosol radiative forcing as our primary metric, which is constrained by the available satellite AOD measurements, and gone to some lengths to show that the relevant variability of this metric is well correlated with MODIS AOD over the time period of our study. Furthermore, we have shown that our results for the relationships of African Easterly Waves to aerosols are consistent whether we use the MERRA-2 radiative forcing metric or the MODIS AOD in the new version of our manuscript **(Figures S3 and S4)**. We believe that this effort has gone some distance to show that our results are robust to uncertainties in the various metrics of aerosol variability. We have not investigated VIIRS data. We are not sure that we agree with the reviewer that it "would not be a challenge using level 2 data for this manuscript". This would be a substantial effort requiring the automated processing with software of terabytes (probably 10s to 100s of terabytes) of level 2 data to capture the temporal and spatial ranges covered in the analysis in this study. While we are not ones to shy away from a challenge, we would argue that the months of work this would entail would not add any additional insight to the analysis we have performed, nor would it add any additional value to the substantial level of effort NASA has already made to aggregate its level 2 data into level 3 data products.

2.

**2 was blanked.**

3. Page 17. Line 17: How did you calculate aerosol shortwave radiative forcing from the MERRA-2? It would be nice to reference Eq. 1

Eq. (1) is mentioned in the revised manuscript to clarify how we calculated aerosol radiative forcing from MERRA-2.

4. Have you compared MERRA-2 AOD with the satellite data? While correlation factors are high and p values are low, the magnitude in the color bars and gradients of the data in the maps are different. I want to remind co-authors that two datasets can have high correlation factors with high biases. My concerns are related to the ocean gradient specifically.

We have compared MERRA-2 aerosol radiative forcing with the MODIS AOD satellite data. This is the subject of figure 2, panels "a" through "e" in the version of the manuscript that was originally submitted. These, of course, are not a direct comparison since the atmospheric radiative forcing of aerosols is not the same quantity as aerosol optical thickness. Nevertheless, the two quantities are highly correlated, as the figure indicates (panels "d" and "e"). It is not necessarily within the scope of this study to validate the MERRA2 AOD against the MODIS AOD. Indeed, there may be bias errors between them, although it is worth noting that MERRA2 assimilates the MODIS AOD, so one must address that lack of independence in any comparison. Nevertheless, as noted above, we have primarily focused on the variations in aerosols and their relation to African Easterly Waves. For this analysis, we believe that the reasonably high correlation between the MERRA-2 radiative forcing and the MODIS AOD is likely both a result of the assimilation of MODIS AOD in the MERRA2 products, and a likely explanation for why our results are robust to whether we use the MERRA-2 atmospheric radiative forcing or the MODIS AOD in our analysis.

5. Page 19. Lines 1-3. I encourage you to show these results in an appendix.

**Hovmöller analysis** using MODIS data are added to the appendix (**Figure S3**) in the new version of the manuscript.
We also conducted **composite analysis** using MODIS AOD, shown in Appendix (**Figure S4**).
We mentioned this in the beginning of the last paragraph of Section 3.3.1 that "*We conducted the same composite analysis using MODIS AOD, which shows that the results are consistent whether the MERRA-2 radiative forcing metric or the MODIS AOD data are applied (Figure S4).*

6. Section 3.3 states previous studies have discussed the development of the AEWs, but I did not find any reference or comparison to other studies in sections 3.3.1 and 3.3.2.

The reviewer is correct to suggest that we should have made more effort to explain how our results relate to some of the prior literature on African Easterly Waves. Rather than do this in section 3, where we report the details of the results of our study, however, we have chosen to do this in the **Conclusion Section** of the paper, where it seemed more natural to summarize the results and reflect on where they fit within the context of the prior literature.

**Conclusions**
Conclusions in this manuscript are not reinforced by any peer-review publication.

The Following parts are added under Conclusion Section in the revised version of the manuscript to address our study with previous peer-reviewed studies:

*"While previous studies showed the relationship between dust transport and AEJ and AEWs across the Atlantic Ocean (Perry et al., 1997; Liu et al., 2008; Francis et al., 2020; Francis et al., 2021) the feedback of dust to AEJ-AEW is not well understood. A few recent studies showed that dust affects the atmospheric dynamics of the Atlantic Ocean by enhancing AEJ and AEW strength (e.g., Bercos-Hickey et al., 2017; Grogan et al., 2016). However, the mechanisms of such effects are still open questions. "*

*"This agrees with in-situ measurements (Soupiona et al., 2020) and regional climate modeling (Saidou Chaibou et al., 2020) of Saharan dust radiative forcing."*

*"The dust-induced enhancement of AEW through a buoyancy source was shown by Grogan et al. (2016), albeit with a different methodology (i.e., analytical and regional modeling analyses). In addition, our results agree with a case study of the Saharan dust event by a regional climate model (Bercos-Hickey et al., 2017) that showed that Saharan dust causes AEW to shift northward and expand westward."*

**Data availability**
Page 26. Line 9: Use MODIS AOD retrievals instead of observations. This study did not use direct observations from MODIS.

This is fixed in the revised manuscript.

**Figures**
Figure 2.a. Did you mean MODIS Dark-Target AOD (550 nm)? Did you use 500 nm instead of 550 nm? https://atmosphere-imager.gsfc.nasa.gov/sites/default/files/ModAtmo/L3_ATBD_C6_C61_2019_02_20.pdf

There was a typo in the figure title: 500 nm is changed to 550 nm.

Figure 2.b. Did you mean MODIS Deep-Blue AOD (550 nm). I encourage consistency in figs 2 a and b.
The typo about the wavelength has been fixed in figure 2 as well.

---

## Author Response (AR2)

The major concerns raised by the 2 reviewers were not addressed adequately. I would like the authors to focus on these two comments without avoiding them:

The comment of reviewer #1 concerning the following caveat: "This differentiation between dust-laden and dust-free analysis is important to bring out the effects of dust on a dynamical system that is strongly coupled to its variability". I expect a more complete answer than simply mentioning that this will be studied in a future paper.

We appreciate the editor suggesting that further effort is required to respond to the comment of reviewer #1 about diagnosing the response to the dust in a dynamical system where the dust is strongly coupled to the variability of the dynamics. First, we have performed the additional analysis suggested by the reviewer to evaluate our results for different wind speed conditions of the African Easterly Jet. This was accomplished by reproducing the result shown in Figure 4a of the paper separately for samples from 3 separate terciles of the AEJ mean wind speed; i.e. the samples associated with the lower, middle, and upper third of mean AEJ wind speed. There are some subtle differences in the resulting relationships between the AEW EKE and the dust radiative forcing, but the main result, that the EKE is enhanced along the southern track of AEWs to the south of the core of the AEJ and in the outflow region to the west of the northern track of AEWs, is robust to variations in the AEJ speed. This at least builds some confidence that the result we are seeing is not just a consequence of a simple correlation of both components to variations in the AEJ dynamics.

We added this result to the supplementary figures (now Figure S5). Discussion of these results, including the point made above, is now included in the manuscript on Page 24 paragraph beginning Line 5 through Line 16 in the version of the revised manuscript with tracked changes.

Furthermore, we have expanded the discussion in Section 2.1 (Page 8 paragraph beginning Line 10 in the version of the revised manuscript with tracked changes and Section 4 (Conclusion; Page 29 paragraph beginning Line 16 to the end of the paragraph on Page 30 Line 22, to expand upon this point and where this paper, given the methodological choices made, fits within the broader study of the role of dust effects on AEW dynamics. We have tried to be more explicit and direct about the limitations of using reanalysis data and the fact that the dust and the waves are fully coupled to the circulation in this dataset. Nevertheless, we argue that the use of a model more tightly constrained to observations, and hence exhibiting a more realistic representation of the circulation, is an advantage of this study. We acknowledge that there exists a prior body of both empirical and modeling work that has advanced the hypothesis that dust radiative effects may be playing a role in enhancing AEW activity and that our empirical approach seeks to determine if the empirical relationships in reanalysis data are consistent with that hypothesis. We argue that both the addition of the analysis suggested by the reviewer (discussed above), as well as the temporal lag analysis that was included in the original submission, are included precisely to help us build confidence that the empirical relationships are consistent with the hypothesis and not just a consequence of a correlation of both dust and AEWs with the overall speed of the AEJ circulation.

Furthermore, we seek to advance a methodology for exploring the energetics of AEWs and their relation to dust radiative effects that is suitable for probing in greater detail how dust may affect

the energetics of AEWs and is a methodology that can be applied equally to reanalysis output, as well as the output from controlled model experiments where dust radiative effects can be turned on and off. We believe that the work presented in this manuscript is a reasonable addition to the growing body of research on this topic. Also, it advances a novel approach to studying this problem, and helps to evaluate an open hypothesis in the literature for the effect of dust radiative processes on AEW dynamics.

As reviewer #2 points out, a discussion of the results described is absent from the manuscript: "There is no discussion about their results (especially section 3 and conclusions) with updated research (For example, the above references). The manuscript needs to be improved and justified through discussions of novel peer-reviewed publications." The few sentences that were added to the conclusion do not constitute a discussion and there should be more effort put into showing how the results from this work complement what other authors have published before.

We appreciate the editor suggesting that further effort is required to respond to the comment of reviewer #2 about the discussion of the results with updated research.

1. We added a new table (Table 4) to the Conclusion Section to summarize the important studies with similar topics (Page 42 in the version of the revised manuscript with tracked changes):

**Table 4.** Summary of relevant publications focused on the impact of dust on AEJ/AEWs.

| Study type | Publication | Highlights |
|---|---|---|
| Data analysis | Jones et al. (2003; 2004) | Using 22-year reanalysis data and the outputs of a dust model, they showed that dust is associated with the enhancement of AEWs. |
| | Hosseinpour and Wilcox (2014) | Using 13-year reanalysis and satellite data, they showed that dust radiative forcing is correlated with meteorological features of AEWs. |
| Modeling | Ma et al. (2012) | By conducting regional numerical simulations of WRF for dust outbreaks and modifying heating rates within the model as a way to account for dust, they showed that dust heating has a weak positive impact on AEWs via promoting convection. |
| | Grogan et al., (2016; 2019) | Using an idealized version of WRF coupled with a dust model and with a supercritical background flow, they found that dust enhances AEWs through a buoyancy source. |
| | Bercos-Hickey et al. (2017; 2020) | They performed numerical simulations using WRF radiatively coupled with a dust model, and showed that both AEJ/AEWs shift northward and westward by dust. |

2. We added more information to the first paragraph of the Conclusion Section, Page 27 beginning Line 19 to Page 28 Line 8 in the version of the revised manuscript with tracked changes:

"While previous studies showed the impact of AEJ Saharan dust transport across the Atlantic Ocean (Perry et al., 1997; Liu et al., 2008; Francis et al., 2020; Francis et al., 2021) the feedback of dust to AEJ-AEW is not well understood. A few recent studies showed that dust affects the atmospheric dynamics of the Atlantic Ocean by enhancing AEW strength (e.g., Jones et al., 2003; 2004; Ma et al., 2012; Hosseinpour and Wilcox,

2014; Grogan et al., 2016; 2019; Bercos-Hickey et al., 2017; 2020) (Table 4 is provided for more details). However, the mechanisms of such effects are still unclear. Moreover, to the best of our knowledge, the mechanistic effects of dust on the eddy energetics of the waves have not been addressed in previous studies. This has motivated us to explore relationships between dust outbreaks and metrics that quantify the production of eddy kinetic energy in AEWs toward a deeper understanding of the role that the dust radiative effect may play in the production of eddy kinetic energy of AEWs."

3. We also added the following part in the last paragraph of the Conclusion Section (Page 30 Lines 16 to 22 in the version of the revised manuscript with tracked changes:

"Although a few studies (e.g., Bercos-Hickey et al., 2017; 2020) used regional models, to the best of our knowledge, there is no global climate model study that explicitly quantifies the impact of dust on AEWs in a coupled system. The empirical relationships apparent from this study will be examined in a follow-on study of atmospheric general circulation model simulations using the Community Earth System Model (CESM) with and without the dust radiative effect to further explore the hypothesis linking dust radiative effects to AEW dynamics."

The following parts of the manuscript also address the reviewer's comment. These were already included in the previous version of the manuscript:

4. Objectives of our study, compared to previous studies- Last paragraph of the Introduction Section; Page 6 Lines 9- 21 in the version of the revised manuscript with tracked changes:

"While previous studies showed the impacts of dust aerosols on climate (Ming and Ramaswamy, 2011; Hosseinpour and Wilcox, 2014; Chen et al., 2021; Liang et al., 2021; Grogan et al., 2022), hydrological cycle (Konare et al., 2005; Kim et al., 2010; Bercos-Hickey et al., 2020) and cloud properties (Weinzierl et al., 2017; Haarig et al., 2019), these elements of the climate system in this region exhibit strong variability due to AEWs. To understand the details of interactions between dust aerosols and climate over the Atlantic Ocean, it is essential to understand how the evolution of AEWs is determined by both diabatic heating, as well as exchanges of eddy kinetic energy (EKE) within the jet-wave system and how dust may contribute to the energy driving AEWs. Toward this goal, we apply eddy energetic concepts to further analyze the relationships between dust and the AEJ-AEWs system to gain insight into the impacts of the dust aerosol radiative effects on the development of AEWs and the distribution of kinetic energy from the source of instability (i.e., AEJ)."

5. Advantage of our method, compared to the methodology of the previous studies- Section 3, the first paragraph of Summary of the results; Page 15 Line 6-19 in the version of the revised manuscript with tracked changes:

"Traditional studies have used the mid-tropospheric trough and ridge from unfiltered wind fields to diagnose the AEWs. In this manner, the AEWs trough was identified where the meridional wind at the vertical level of the AEJ is equal to zero, indicating that the wind shifts from northerlies to southerlies (Diedhiou et al., 1999). The existence of two distinct tracks of the AEWs: the northern and southern tracks (e.g., Diedhiou et al.,

1999; Nitta and Takayabu, 1985; Reed et al., 1988; Wu et al., 2013) have been identified by examining the vorticity structure of the AEWs (e.g., Carlson 1969 a&b; Thorncroft and Hodges, 2001; Hopsch et al., 2007) and applying the reversal of the meridional gradient of potential vorticity (e.g., Norquist et al., 1977; Pytharoulis and Thorncroft, 1999; Kiladis et al., 2006). However, these methods are limited because of the overlapping scale of AEWs with other phenomena and the significant amount of manual intervention required to differentiate between synoptic-scale AEW trough axes and localized circulation centers. As a solution to this problem, here we applied the eddy energy budget to diagnose the growth and evolution of the AEWs."

6. Comparison of our results with the previous studies- Section 3, Summary of the results, Page 16 Lines 15-18 in the version of the revised manuscript with tracked changes:
   "These are consistent with the previous studies, showing that after leaving the West coast of Africa, the majority of AEWs either (1) penetrate the subtropical Atlantic Ocean via an interaction with an extratropical trough, or (2) develop further downstream and are involved in tropical cyclogenesis (Berry et al., 2007; Chen et al., 2008)."

7. Comparison of our result with the previous studies- Conclusion Section, Page 29, Lines 4- 8 in the version of the revised manuscript with tracked changes:
   "The dust-induced enhancement of AEW through a buoyancy source was shown by Grogan et al. (2016), albeit with a different methodology (i.e., analytical and regional modeling analyses). In addition, our results agree with a case study of the Saharan dust event by a regional climate model (Bercos-Hickey et al., 2017) that showed that Saharan dust causes AEW to shift northward and expand westward."

8. Since reviewer#2 provided a full review prior to the Discussion Process, we took diligent efforts to address all the reviewer's comments in the version of the manuscript we submitted earlier for the Online Discussion Process. For instance, all the references that reviewer#2 suggested above were included in the manuscript:
   1.Francis et al. (2021) study has been added on Page 4 Line 11 in the version of the revised manuscript with tracked changes.
   2.Meloni et al., (2018) study has been added on Page 4 Line in the version of the revised manuscript with tracked changes.
   3.Bercos-Hickey et al., (2017) study has been referenced several times in the version of the revised manuscript with tracked changes, including Page 5 Line 3, Page 29 Line 10, and Page 30 Line 16.
   5.Grogan et al., (2017) study has been added on Page 29 Line 7 and Page 42 (Table 4) in the version of the revised manuscript with tracked changes.
   6.Grogan et al., (2019) study has been added on Page 4 Line 21, Page 28 Line 2, and Page 42 (Table 4) in the version of the revised manuscript with tracked changes.
   7.Bercos-Hickey et al., (2019) study has been added on Page 4 Line 21 and Page 42 (Table 4) in the version of the revised manuscript with tracked changes.
   8.Francis et al. (2020) study has been added on Page 27 Line 21 in the version of the revised manuscript with tracked changes.